# Predicting memorization within Large Language Models fine-tuned for classification

## Abstract

Large Language Models have received significant attention due to their abilities to solve a wide range of complex tasks. However these models memorize a significant proportion of their training data, posing a serious threat when disclosed at inference time. To mitigate this unintended memorization, it is crucial to understand what elements are memorized and why. Most existing works provide *a posteriori* explanations, which has a limited interest in practice. To address this gap, we propose a new approach to detect memorized samples *a priori* in LLMs fine-tuned on classification tasks. This method is efficient from the early stages of training and readily adaptable to other classification settings, such as training vision models from scratch. Our method is supported by new theoretical results that we demonstrate, and requires a low computational budget. We obtain strong empirical results, paving the way for systematic inspection and protection of these vulnerable samples before memorization happens.

## 1 Introduction

Large Language Models (LLMs) have revolutionized the way we approach natural language understanding. The availability to the general public of models such as ChatGPT, capable of solving a wide range of tasks without adaptation, has democratized their use. However, a growing body of research have shown that these models memorize a significant proportion of their training data, raising legal and ethical challenges (Zhang et al., 2017; Carlini et al., 2023; Mireshghallah et al., 2022b). The impact of memorization is ambiguous. On the one hand, it poses a serious threat to privacy and intellectual property because LLMs are often trained on large datasets including sensitive and private information. Practical attacks have been developed to extract this information from training datasets (Carlini et al., 2021; Lukas et al., 2023; Yu et al., 2023; Nasr et al., 2023), and LLMs have also been shown to plagiarize copyrighted content at inference time (Lee et al., 2023; Henderson et al., 2024). On the other hand, memorization can positively impact model's performance, because memorized samples are highly informative. Studies have revealed that outliers are more likely to be memorized, and that these memorized outliers help the model generalize to similar inputs (Feldman, 2020; Feldman & Zhang, 2020; Wang et al., 2024).

Mitigating the negative impacts of memorization while still harnessing its advantages is a challenging task, that requires varying approaches based on the sensitivity of the training data and the purpose of the model. However, practitioners often struggle to evaluate the potential risk of memorized samples, as empirical defenses often fail to capture the most vulnerable samples from the training set (Aerni et al., 2024). To address this limitation, we propose a new method to audit models under development and predict, from the early stages of training, the elements of the training data that the LLM is likely to memorize. Our first goal is to provide practitioners with an efficient tool to inspect vulnerable elements and select an appropriate mitigation strategy: anonymization, differential privacy, acceptance of the risk, etc. Our second goal is to enable researchers to design new empirical defenses that optimally allocate their privacy budgets to protect the most vulnerable samples, thereby achieving a better privacy-utility trade-off. For both goals, it is crucial to predict memorization early in the training pipeline and at minimal cost. Indeed, *a-posteriori* measures of memorization, such as LiRA (Carlini et al., 2022a) or counterfactual memorization (Feldman & Zhang, 2020), require not only the completion of the training of the model, but also the training of several *shadow models*, making them prohibitively computationally expensive for most practitioners. On the other hand, for researcher developing empirical defenses, it is crucial to detect vulnerable

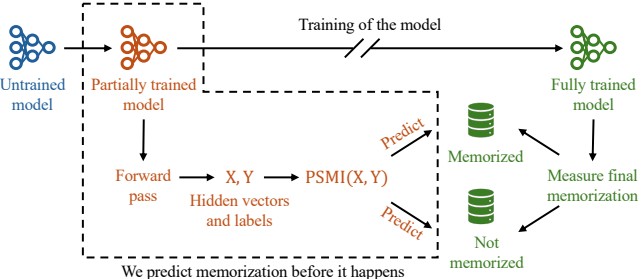 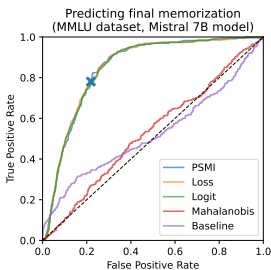

(a) We predict memorization from the early stages of training    (b) Performance of the prediction

Figure 1: Figure 1a: We interrupt training when the median training loss has decreased by 95%. We compute a forward pass to retrieve $X$, the hidden representation of the inputs within the partially trained model. We measure the consistency between $X$ and the label $Y$, and use it to predict memorization within the fully trained model. Figure 1b: Evaluation of the four metrics we used to quantify the consistency between $X$ and $Y$: PSMI, loss, logit gap, and Mahalanobis distance. "Early memo" is our baseline, adapted from Biderman et al. (2023) (see Appendix C.1). The cross represents the default threshold for PSMI, equal to zero (see Algorithm 1 and Theorem 1).

samples as soon as possible to protect them before they are memorized. Our method achieves this by predicting memorization after only a fraction of an epoch, without requiring any shadow model.

To predict memorization before it occurs, we interrupt training when the median training loss has significantly decreased, typically by 95% (see Figure 1). This drop indicates that the model has learned simple patterns in the hidden representations, enabling it to accurately classify the majority of samples, without relying on memorization. At this stage, we measure the consistency between the labels and the hidden representations of the elements within the partially trained model. If a hidden representation fails to adequately explain its assigned label, it indicates that the data sample behaves as a local outlier, within the data distribution's long tail (Zhu et al., 2014). Such outliers are particularly vulnerable to memorization, because the model will likely fail to learn meaningful representations for them, and will instead resort to memorizing them (Feldman & Zhang, 2020).

We evaluated four approaches to quantify the consistency between the hidden representations and the labels, with the objective of predicting memorization in the fully trained model: loss, logit gap, Mahalanobis distance (Mahalanobis, 1936), and Pointwise Sliced Mutual Information (PSMI) (Goldfeld & Greenewald, 2021; Wongso et al., 2023a). With the exception of Mahalanobis distance, all approaches achieved strong empirical results. The loss is straightforward to implement and fast to compute, but requires an additional hyperparameter to define a threshold for separating elements predicted to be memorized. The logit gap offers no advantage over the loss. On the other hand, PSMI saves one hyperparameter because we demonstrated that zero is a natural threshold to use. However, it marginally increases computational cost and is more complex to implement.

To the best of our knowledge, Biderman et al. (2023) provides the only baseline to which our approach can be compared. They predict memorization in LLMs trained on generative tasks, with a reasonable computational budget and *before* the end of training. However, memorization is defined differently for generative and discriminative tasks. They use *k-extractability* (Carlini et al., 2021), which is very cheap to compute for generative models, but not applicable to classification models. For these models, memorization is typically defined as vulnerability to membership inference attack (Shokri et al., 2017), which is more computationally expensive. Our approach is only applicable to classification models, for which we found no directly comparable baseline. This is why we adapted the method of Biderman et al. (2023) to a classification setting, despite the prohibitive computational cost arising from the increased complexity of measuring memorization (see Appendix C.1). Even with this adaptation, we observed similar results: at the early stages of training, it is possible to achieve a low False Positive Rate (FPR), but not a high True Positive Rate (TPR), because vulnerable samples have not yet been memorized. Conversely, our approach obtains both high TPR and low FPR, paving the way for inspecting and protecting vulnerable samples under realistic conditions.

**Our main contributions can be summarized as follows.**

- We demonstrate that it is possible to predict, from the early stages of training, whether a sample will be memorized when fine-tuning a LLM for a classification task;
- We formalize the threat model and propose FPR at high TPR as the evaluation metric;
- We compare several metrics and discuss their respective advantages;
- We validate the effectiveness of our approach for three different 7B LLMs fine-tuned on three distinct multi-choice question datasets;
- We demonstrate its adaptability by applying it as-is to vision models trained from scratch.

## 1.1 RELATED WORK

**Membership Inference Attacks (MIA)**   These attacks were first introduced by Shokri et al. (2017), and aim to determine whether a target individual element was part of a target model's training set. Although they are less realistic and practical than extraction attacks (Carlini et al., 2021; Lukas et al., 2023; Nasr et al., 2023), membership inference attacks have become the standard approach for measuring the amount of private information a model can leak. Popular attacks such as those of Shokri et al. (2017); Carlini et al. (2022a); Wen et al. (2023) involve training a large number of *shadow models* with different training data. Due to the significant computational resources required, alternative attack methods have been developed that necessitate training fewer shadow models or none at all (Yeom et al., 2018; Mattern et al., 2023; Zarifzadeh et al., 2024).

**Several definitions of unintended memorization in neural networks**   For discriminative models, memorization is usually defined as vulnerability to MIA, as in (Mireshghallah et al., 2022a; Carlini et al., 2022b; Aerni et al., 2024). Counterfactual memorization can also be applied to such models, requiring the training of multiple models with varying datasets to capture the influence of individual data samples (Feldman & Zhang, 2020). On the opposite, to focus on more realistic threats, memorization can be defined as vulnerability to extraction or reconstruction attacks (Carlini et al., 2018; 2021; 2023; Biderman et al., 2023; Lukas et al., 2023; Dentan et al., 2024). These definitions are mostly used with generative models, as such attacks are more complex to implement on discriminative models and often achieve lower performance. As pointed out by Lee et al. (2022); Prashanth et al. (2024), a large majority of elements extracted consist of common strings frequently repeated in standard datasets. This is why counterfactual memorization was adapted to generative models (Zhang et al., 2023; Wang et al., 2024; Pappu et al., 2024; Lesci et al., 2024). Finally, MIA can also be used for generative models (Meeus et al., 2024).

**Explaining and predicting memorization**   In machine learning, memorization has been commonly associated with overfitting and considered the opposite of generalization. However, this belief was challenged by Zhang et al. (2017), who proved that a model can simultaneously perfectly fit random labels and real samples. This phenomenon was studied further by Arpit et al. (2017); Chatterjee (2018), followed by Feldman (2020) who provided a theoretical framework to explain how memorization can in fact increase generalization. His idea is that a substantial number of elements in typical datasets belong to the long tail of the distribution (Zhu et al., 2014), meaning that they behave like local outliers that are unrepresentative of the overall distribution. As a result, memorizing these elements enables the model to generalize to similar samples at inference time. This idea was confirmed empirically by Feldman & Zhang (2020), and later by Zhang et al. (2023), who observed that memorized samples are relatively difficult for the model. Similarly, Wang et al. (2024) observed that memorization in self-supervised learning can increase generalization.

A different approach to explain memorization is to analyze the hidden representations learned by the model. For example, Azize & Basu (2024) linked the privacy leakage of a sample to the Mahalanobis distance (Mahalanobis, 1936) between the sample and its data distribution. Leemann et al. (2024) evaluated several metrics to predict memorization from a reference model, and concluded that test loss is the best predictor. Wongso et al. (2023b) computed Sliced Mutual Information (Goldfeld & Greenewald, 2021) between the hidden representations and the labels. They theoretically explain why a low SMI indicates memorization, and successfully observed this phenomenon in practice.

These approaches provide *a posteriori* explanations of memorization, because they are either computed from the fully trained model or from a reference model. On the opposite, Biderman et al.

(2023) introduced a new method to predict memorization *before* the end of pre-training. They achieve promising results with high accuracy. However, they obtain low recall scores, indicating that a significant proportion of the samples that are memorized by the final model cannot be detected using their metrics. As they acknowledge, this is an important shortcoming of their method.

## 1.2 Problem setting

**Threat model: predicting memorization, not mitigating it**   We adapt the setting of Biderman et al. (2023). We assume that an engineer is planning to fine-tune a LLM on a private dataset for a classification task, where a small proportion of the dataset contains sensitive information that should not be memorized by the model for privacy concerns. The engineer has full access to the model, its training pipeline and intermediate checkpoints. They do not have the computational budget to train the shadow models required for *a posteriori* measures of memorization such as LiRA or counterfactual memorization (see Section 1.1). Consequently, they aim to conduct some tests at the beginning of the full training run to approximate *a posteriori* memorization, and determine if the sensitive samples will be memorized by the fully trained model (see Figure 1).

The engineer wishes to dedicate only a small amount of compute for these tests, to reduce the overhead of confidentiality checks. Moreover, they aim to detect vulnerable samples early to inspect them before they are memorized and decide whether to accept the privacy risk, anonymize or remove the samples, or implement mitigation techniques. This is particularly important for researchers developing empirical defense that optimally allocate their privacy budgets to protect the most vulnerable samples without altering non-vulnerable ones, thereby achieving a better privacy-utility trade-off. We make no assumptions about the subsequent decisions made by the engineer, and only focus on developing a good predictor of which elements will be memorized by the final model.

**Evaluation metrics: FPR at high TPR**   We use predictions from on the partially trained model to predict memorization in the fully trained model. As for membership inference attacks, we evaluate the True Positive Rate / False Positive Rate (TPR / FPR) trade-off in the prediction (Carlini et al., 2022a). The TPR represents the proportion of memorized samples in the final model that are correctly detected based on the partially trained one, and the FPR represents the proportion of non-memorized samples that are wrongly detected. We prefer TPR / FPR to precision / recall because it is independent of the prevalence of memorized samples. However, as noted by Biderman et al. (2023), a high TPR is more important than a low FPR. Indeed, false positives lead the engineer to be overly cautious, which is unprofitable, but does not threaten privacy. Conversely, false negatives lead the engineer to underestimate memorization, which entails a privacy risk. As a consequence, we will focus on regions of the TPR / FPR curves that achieve a high TPR, typically greater than 75%. The Area Under the Curve (AUC) provides a single numerical value for comparing metrics, although it presents a simplistic view of the TPR / FPR trade-off.

**Experimental settings**   Most studies on memorization in classification settings focus on models of intermediate size trained on datasets such as CIFAR-10 or CIFAR-100 (Aerni et al., 2024; Carlini et al., 2022b; Feldman & Zhang, 2020). We have decided to consider more recent scenarios using LLMs fine-tuned for classification tasks. Indeed, generative models are increasingly trained to produce formatted outputs for tasks previously handled by discriminative models, such as information extraction (Kim et al., 2022; Dhouib et al., 2023), sentiment analysis (Šmíd et al., 2024), or recommendation (Geng et al., 2022; Cui et al., 2022). Moreover, privacy is often a significant concern for fine-tuning, as the datasets used for this purpose frequently contain sensitive private information.

Although our experiments focus on fine-tuned LLMs, our method relies on the specific properties of neither LLMs nor fine-tuning. Consequently, our method is suitable for any model trained for classification tasks. In Section 3.2, we apply our method as-is to a Wide Residual Network (Zagoruyko & Komodakis, 2016) trained from scratch on CIFAR-10, yielding conclusive results.

For most experiments, we used three pretrained models with similar architectures: Mistral 7B v1 (Jiang et al., 2023), Llama 7B v2 (Touvron et al., 2023), and Gemma 7B (Team et al., 2024). We used three popular academic benchmarks: MMLU (Hendrycks et al., 2021b), ETHICS (Hendrycks et al., 2021a) and ARC (Boratko et al., 2018). We fine-tuned these models using LoRA (Hu et al., 2022) and question-answering templates asking the model to output the label. Models are trained using Next Token Prediction task, computing the loss only for the token corresponding to the label.

## 2 METHODOLOGY

### 2.1 PRELIMINARY

**Hidden representations in Large Language Models**  We consider a decoder-only transformer-based LLM such as Llama 2 (Touvron et al., 2023) trained on a multi-choice question (MCQ) dataset such as MMLU (Hendrycks et al., 2021b). With this type of architecture, all tokens of the input are embedded into *hidden representations* in $\mathbb{R}^d$. They are successively transformed at each of the $K$ layers to incorporate information from the context. For example, Llama 2 7B uses $d = 4096$ and $K = 32$. Finally, the representation of the last token at the last layer is used to predict the answer.

For $k \in [\![1, K]\!]$, let $X_k \in \mathbb{R}^d$ be the hidden representation of the last token after the $k$-th layer, and $Y \in \{0, 1, 2, \ldots, r\}$ the answer of the MCQ. We can think of $X_k$ and $Y$ as random variables following a joint probability distribution $\mathcal{D}_k$ that can be estimated from the dataset. In the following, we use information-theoretic tools to analyze the interplay between variables $X_k$ and $Y$. Note that $\mathcal{D}_k$ and $X_k$ depend of the training step, but we omit this aspect in our notations to consider a LLM that we freeze to analyze its representations.

**(Pointwise) Sliced Mutual Information**  Sliced Mutual Information (SMI) was introduced by Goldfeld & Greenewald (2021). Similar to Shannon's Mutual Information (denoted I), it measures the statistical dependence between two random variables such as $X_k$ and $Y$. Intuitively, it measures how much the realization of $X_k$ tells us about the realization of $Y$. If they are independent, the mutual information is zero ; and if $X_k$ fully determines $Y$, the mutual information is maximal. In our setting, it represents how useful the hidden representations are to predict the labels. Thus, we expect the SMI to increase with $k$ as the representations become more efficient over layers. Indeed, SMI is not subject to the data processing inequality, contrary to I (Goldfeld & Greenewald, 2021).

**Definition 1** *Sliced Mutual Information (*SMI*) is the expectation of Mutual Information (denoted* I*) over one-dimensional projections sampled uniformly at random on the unit sphere (denoted* $\mathcal{U}(\mathbb{S}^d)$*):*

$$\text{SMI}(X_k, Y) = \underset{\theta \sim \mathcal{U}(\mathbb{S}^d)}{\mathbb{E}} \left[ \text{I}(\theta^T X_k, Y) \right] = \underset{\theta \sim \mathcal{U}(\mathbb{S}^d)}{\mathbb{E}} \left[ \underset{(X_k, Y) \sim \mathcal{D}_k}{\mathbb{E}} \left[ \log \frac{p(\theta^T X_k, Y)}{p(\theta^T X_k) p(Y)} \right] \right] \quad (1)$$

Pointwise Sliced Mutual Information (PSMI) was introduced by Wongso et al. (2023a) and used as an explainability tool. For every individual realization $(x_k, y)$ of the variables $(X_k, Y)$, it represents how surprising it is to observe $x_k$ and $y$ together. For example, a low PSMI means that label $y$ was unexpected with representation $x_k$, maybe because all similar representations to $x_k$ are associated with another $y' \neq y$ in the dataset.

**Definition 2** *Pointwise Sliced Mutual Information (*PSMI*) is defined for every realization $(x_k, y) \in \mathbb{R}^d \times [\![0; r]\!]$ of the variables $(X_k, Y)$ as:*

$$\text{PSMI}(x_k, y) = \underset{\theta \sim \mathcal{U}(\mathbb{S}^d)}{\mathbb{E}} \left[ \log \frac{p(\theta^T x_k, y)}{p(\theta^T x_k) p(y)} \right] \quad (2)$$

Here, $p$ represents the value of the probability distribution function. It depends on the joint distribution $\mathcal{D}_k$, and can be estimated numerically by approximating $p(\theta^T x_k \mid y)$ by a Gaussian (Wongso et al., 2023a). The resulting estimator of PSMI is very fast to compute and easy to parallelize. The bottleneck is to compute the hidden representations $x_k$, which requires one forward pass per sample.

### 2.2 WHY ELEMENTS WITH LOW PSMI ARE LIKELY TO BE MEMORIZED

Intuitively, PSMI measures the dependency between the hidden representation of a data sample and its label. As a result, PSMI should be lower for outliers and points that are hard to classify. Following the results of Feldman & Zhang (2020), these are the points that are most likely to be memorized.

The following theorem validates this intuition. We consider a binary classification setting with balanced classes and some outliers. With probability $1 - \varepsilon$, the point is not an outlier, and the hidden

representation $X$ follows a Gaussian distribution (Eq. 3). This Gaussian behavior is a classical hypothesis derived from the central limit theorem applied to deep neural networks (Matthews et al., 2018). Conversely, with probability $\varepsilon$, the point is an outlier: $X$ does not necessarily follow the Gaussian distributions, and $Y$ is sampled uniformly at random (Eq. 4). We prove that on average PSMI is positive for non-outliers (Eq. 5), and zero for outliers (Eq. 6). See proof in Appendix B.

**Theorem 1** *Let $(X, Y) \in \mathbb{R}^d \times \{0, 1\}$ be random variables. We assume that $p(Y = 0) = p(Y = 1) = 0.5$ and that $X$ is a continuous random variable. We also assume that there exist $\mu_0, \mu_1 \in \mathbb{R}^d$ with $\mu_0 \neq \mu_1$, and $\Sigma_0, \Sigma_1 \in \mathbb{R}^{d \times d}$, and a Bernoulli variable $\Delta \sim \mathcal{B}(\varepsilon)$ with $\varepsilon \in ]0, 1[$ such that:*

$$p(X \mid Y = 0, \Delta = 0) \sim \mathcal{N}(\mu_0, \Sigma_0) \quad \text{and} \quad p(X \mid Y = 1, \Delta = 0) \sim \mathcal{N}(\mu_1, \Sigma_1) \tag{3}$$

$$\forall x \in \mathbb{R}^d, \quad p(Y = 0 \mid \Delta = 1, X = x) = p(Y = 1 \mid \Delta = 1, X = x) = 0.5 \tag{4}$$

*Given this, we then have:*

$$\mathbb{E}_{X,Y} [\text{PSMI}(X, Y) \mid \Delta = 0] > 0 \tag{5}$$

$$\mathbb{E}_{X,Y} [\text{PSMI}(X, Y) \mid \Delta = 1] = 0 \tag{6}$$

### 2.3 OUR METHOD

Based on Theorem 1, we propose Algorithm 1 to predict memorization. The three hyperparameters in bold performed well in every setting we evaluated. We interrupt training when the median training loss decreases by **95%**, as this metric remains stable even in the presence of outliers. We measure PSMI at the **last layer**, which is consistently informative, and use a threshold of **zero**, as supported by Theorem 1. These default values yielded conclusive results when applied to CIFAR-10, for which they were not optimized (see ablation studies in Section 3.2). Consequently, these hyperparameters are likely suitable for practitioners auditing models in diverse classification settings. To facilitate the use of this method, we provide a PyPI package containing an automated estimator of PSMI.[1]

As noted in the introduction, using the loss instead of PSMI also produced convincing results. An alternative to Algorithm 1 is to replace lines 2-3 with a forward pass to retrieve the loss (see Algorithm 2). The implementation is simpler, but it requires the practitioner to select a threshold to separate samples predicted to be memorized, as there is no natural threshold like zero for the PSMI.

---

**Algorithm 1** Using PSMI to predict memorization

---

1: Interrupt training when the median training loss has decreased by at least **95%**.
2: Compute a forward pass for every sample to retrieve the hidden vector after the **last layer**.
3: Use Algorithm 1 in (Wongso et al., 2023a) to estimate PSMI for every sample.
4: Predict that every sample with **PSMI $\leq$ 0** will be memorized.

---

## 3 EXPERIMENTAL RESULTS

We evaluate the efficiency of predicting memorization from the early stages of training, using several metrics that quantify the consistency between the hidden representations and the assigned label: PSMI (Algorithm 1), loss, logit gap, and Mahalanobis distance (see Appendix C.3). We compare these predictors to the baseline of Biderman et al. (2023), which we adapted to our classification setting. While its computational cost is much higher, it remains the only comparable approach we are aware of (see Appendix C.1). We use five combinations of dataset/model: ARC/Mistral, ETHICS/Mistral, MMLU/Mistral, MMLU/Llama and MMLU/Gemma (see Section 1.2).

To evaluate our approach, we resume training and measure memorization at the end. To ensure a fair comparison between our experiments and prevent over-training, we always stop training after

---

[1]`hidden_github_url_pypi_package_for_review`

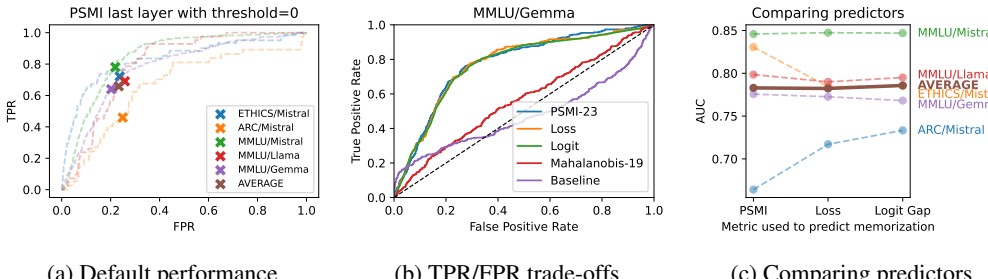

(a) Default performance  (b) TPR/FPR trade-offs  (c) Comparing predictors

Figure 2: Figure 2a: TPR/FPR trade-off of PSMI using the default hyperparameters of Algorithm 1 (crosses) compared to the trade-offs that can be obtained with the best layer (dashed lines). The prediction is computed when the median training loss has decreased by 95%. Figure 2b: Our baseline and Mahalanobis distance have near-random performance, whereas PSMI, loss, and logit gap are good predictors. "23" and "19" denote the layers used for computation, which perform best in these settings. Figure 2c: Comparing the AUC of the best predictors.

one epoch. As in (Carlini et al., 2022b; Mireshghallah et al., 2022b; Aerni et al., 2024), we use vulnerability to LiRA membership inference attack as our ground truth memorization metric (see Appendix A.1). This attack provides a numeric score for each sample, which is a likelihood ratio computed from a large number of *shadow models*. We always display the natural logarithm of LiRA, so a positive score indicates that the element was memorized. Unless otherwise stated, memorized samples are defined as those with log-LiRA $\geq 4$, which corresponds to LiRA $\gtrsim 54.6$.

**Computational gains**  The bottleneck of Algorithm 1 is computing a forward pass for every sample, which costs as much as $1/3$ of an epoch (Hobbhahn & Jsevillamol, 2021). Moreover, we typically compute our metric after only 0.2 to 0.4 epochs (see Section 3.2). Thus, our method costs about as much as $2/3$ of an epoch. On the opposite, our ground truth memorization requires training 100 models for one epoch, which is 150 times more expensive. As explained in Section 1.2, practitioners within our threat model do not have the budgets to compute such *a posteriori* measures of memorization. Our approach enables them to approximate memorization at minimal cost.

We used a HPC cluster with Nvidia A100 80G GPUs and Intel Xeon 6248 40-cores CPUs. The total computational cost of our experiments is 10961 GPU hours and 5787 single-core CPU hours. This represents 0.57 tCO$_2$eq for this cluster (see `hidden_hpc_url_for_review`).

### 3.1 MEMORIZATION CAN BE RELIABLY PREDICTED

Our experiments demonstrate that memorization can be predicted accurately from the early stages of training. In Figure 2a, we present the TPR and FPR values achieved with the procedure and the default hyperparameters provided in Algorithm 1. On average, we obtained a **FPR of 23.3%** and a **TPR of 65.8%**. These excellent scores prove that most memorized samples can be detected very early (high TPR) and with a great exactness (low FPR).

The crosses corresponding to the default procedure are not exactly on the dashed lines of the same color. This is because the dashed lined are obtained by optimizing both the layer used to compute PSMI and the threshold used to separate samples predicted to be memorized. The proximity of the crosses to the dashed lines indicates that the performance improvement gained from optimizing the layer is minimal (see Section 3.2 for more details).

**TPR/FPR trade-off when optimizing the thresholds (Figures 2b and 2c)**  We vary the threshold used to separate samples predicted to be memorized, resulting in the TPR/FPR trade-off presented in Figure 2b. Our baseline and Mahalanobis distance proved ineffective. On the opposite, the FPR@TPR=75% is equal to 24.1% for PSMI, 24.6% for the loss and 23.1% for the logit gap, which demonstrates that a practitioner can detect the majority of memorized samples with an acceptable FPR. In Figure 2c we observe that PSMI, loss and logit gap perform similarly, and achieve a very high AUC values on average. This demonstrates that they accurately capture susceptible samples from the early stages of training. See Appendix D.1 for additional results.

## 3.2 ABLATION STUDIES

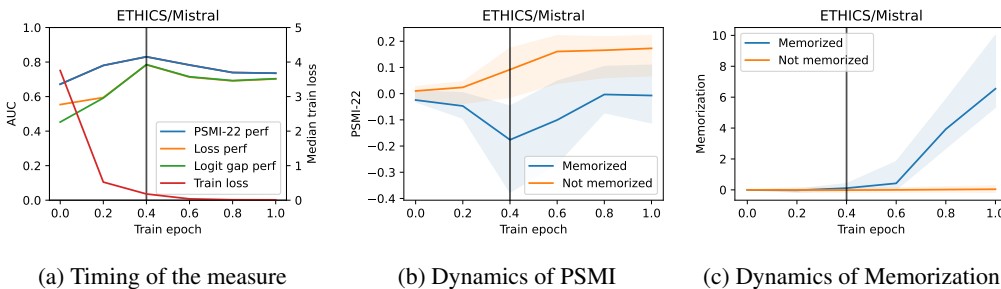

(a) Timing of the measure      (b) Dynamics of PSMI      (c) Dynamics of Memorization

Figure 3: Memorized samples can be detected from epoch 0.4, though they are not yet memorized. Figure 3a: in blue, orange and green, the AUC of PSMI, Loss and Logit Gap ; in red, the median train loss. The vertical line marks the 95% decrease in training loss. Figure 3b: the solid line shows the median PSMI for memorized and non-memorized samples, while the shaded area represents the 25%-75% quantiles. Figure 3c outlines memorization using a similar representation.

**Impact of the timing of the measure (Figure 3)** In Algorithm 1, we predict memorization when the median training loss has decreased by 95%. To validate this choice empirically, we save the models every 0.2 epochs, and evaluate how efficiently memorization can be predicted at each checkpoint. As we observe in Figure 3a, the predictions begin to be effective only when the median training loss has decreased significantly, and the 95% threshold proved to be effective in all our settings. We observe in Figure 3b that it corresponds to the moment when the PSMI of samples memorized by the final model is much lower than that of non-memorized samples. Indeed, the patterns learned by the model earlier are not relevant enough for PSMI to accurately quantify if a sample will likely be hard to learn. Importantly, as shown in Figure 3c, memorized samples are not yet memorized at that moment. This indicates that a practitioner within our threat model can implement mitigation techniques based on the privacy risks associated with memorized samples without restarting the training process. See Section 1.2 for details on our threat model and see Appendix D.2 for additional plots.

**Impact of the memorization threshold (Figure 4a)** As stated at the beginning of Section 3, memorized samples are defined by default as those with log-LiRA $\geq 4$, which corresponds LiRA $\geq e^4 \simeq 54.6$. This indicates strong memorization, as the attack predicts that these elements are 54.6 times more likely to be members of the dataset than non-member (see details in Appendix A.1.1). For example, with MMLU/Llama, 2.8% of the elements meet this definition after one epoch of training (see Appendix D.3 for results in other settings). In Figure 4a, we vary this threshold and measure the AUC using PSMI, loss and logit gap. We also represent the proportion of memorized samples associated with the threshold. We observe that our method is more effective for most vulnerable

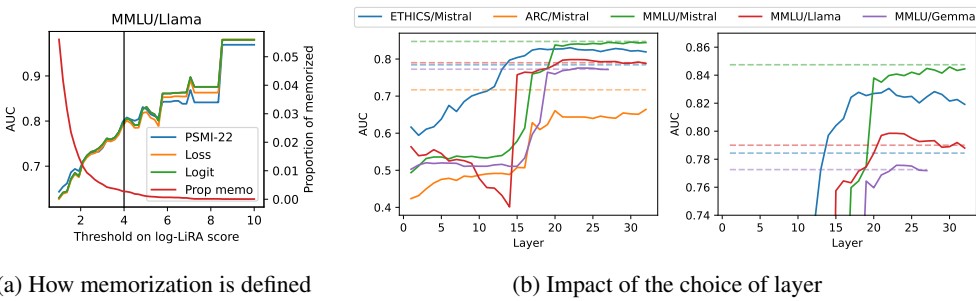

(a) How memorization is defined      (b) Impact of the choice of layer

Figure 4: Figure 4a: impact of the threshold used to define "memorized" and "non-memorized" samples. The vertical bar indicates the default threshold log-LiRA = 4. Figure 4b: impact of the choice of the layer on the performance of PSMI. The solid lines represent the AUC for PSMI at that layer, and the dashed lines represent the AUC for the loss, which does not depend on the layer. The right plot is a zoomed-in view focusing on high AUC regions.

samples, obtaining the highest log-LiRA score. We interpret this as meaning that elements that are clearly detected as memorized by LiRA were necessarily hard to learn for the model, so they can be detected by our method. See Appendix A.2 and Appendix D.4 for additional results and discussions.

**Impact of the layer used to compute PSMI (Figure 4b)**    The default method presented in Algorithm 1 uses hidden representations from the last layer to predict memorization. However, depending on the model and the dataset, different layers can be more effective. We observe that in every setting, only the last layers are useful for predicting memorization. However, it appears that the importance of layers varies depending on the dataset. When we fix the model to Mistral and vary the dataset, we observe that for complex tasks such as MMLU or ARC datasets (with up to 5 possible labels), the curve rises sharply around layers 15–20 and then stabilizes with minor variations. We interpret this to mean that more complex tasks require more intricate interactions between token representations, so relevant layers are concentrated towards the end of the network. On the opposite, for ETHICS dataset, which is a simpler task of binary classification, the curve increases more smoothly. This indicates that samples are easier to separate with fewer interactions between tokens, allowing memorization to be detected from the earliest layers. Conversely, we observed that for a fixed dataset (MMLU), the choice of model has little impact on the shape of the curve. Finally, we observe that across all settings, the difference between the AUC with the last layer and the AUC with the best layer is minor, which justifies selecting the last layer in Algorithm 1.

**Applicability to other classification settings (Figure 5)**    As noted in Section 1.2, our method relies on the specific properties of neither LLMs nor fine-tuning. Consequently, it is suitable for any model trained for classification tasks. To validate this hypothesis, we applied our method as-is to a Wide Residual Network (WRN16-4) (Zagoruyko & Komodakis, 2016) trained from scratch on CIFAR-10. This setting differs significantly from the fine-tuning of LLMs studied so far: the model uses convolutions instead of transformers, is trained on a visual task rather than a textual one, and is trained from scratch rather than fine-tuned. We believe that the excellent performance of our method in this setting indicates that it is applicable to a wide range of classification scenarios.

We adapted the framework of Aerni et al. (2024) to interrupt training and measure the PSMI, loss, and logit gap on a model trained without any mitigation techniques. The authors introduced out-of-distribution *canaries* within the training set, and demonstrated that they correctly mimic the most vulnerable samples. In Figure 5a, we predict memorization using the same definition as above, with a threshold of 4 applied to log-LiRA. Even though our method was applied without any modifications, we obtained very high AUC scores, surpassing those achieved with MMLU/Mistral, which is our best setting for fine-tuned LLMs. However, we note that the default hyperparameters of Algorithm 1 lead to a very good FPR but a low TPR. Indeed, in this setting, 3.8% of samples satisfy log-LiRA $\geq 4$, which is relatively high (see Appendix D.3). In contrast, in Figure 5b, we focus on the most vulnerable samples by predicting the canaries that mimic them. We observe that our method yields excellent results, and that the hyperparameters of Algorithm 1 are well-suited for detecting these

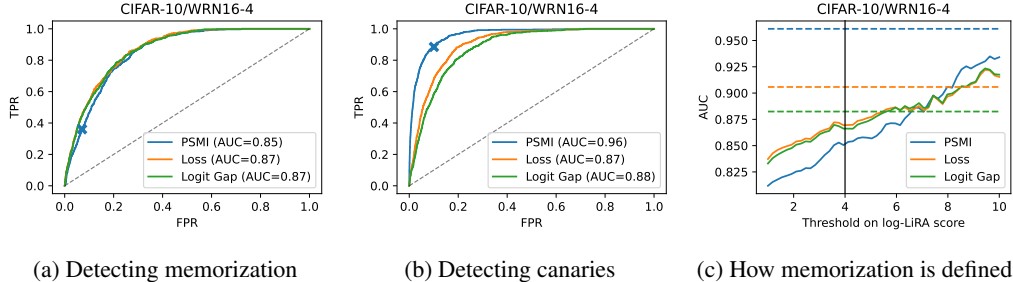

(a) Detecting memorization          (b) Detecting canaries          (c) How memorization is defined

Figure 5: Applying our method as-is on a WRN16-4 trained from scratch on CIFAR-10, adapting the framework of Aerni et al. (2024). Figure 5a: Using PSMI (last layer), loss and logit gap to predict memorized samples. The cross marks the default hyperparameters of Algorithm 1. Figure 5b: Predicting canaries that mimic most memorized samples. Figure 5c The solid line represents the impact of the choice of threshold applied to LiRA to defined "memorized" and "non memorized" samples. The dashed line is the AUC when memorized samples are defined as the canaries.

highly vulnerable samples. In Figure 5c, we confirm that our method obtains better results when detecting samples that are very well memorized, with a high log-LiRA score (see Appendix D.6 for more details).

## 4 FINAL REMARKS

**Ethical considerations** This paper discusses vulnerability to privacy attacks against language models in practical settings, raising ethical considerations due to similar models trained on private data already being attacked in production (Nasr et al., 2023). However, we believe that our work is unlikely to benefit adversaries with harmful intent, for several reasons. First, our approach necessitates access to the checkpoint of a partially trained model, and to the training dataset. In practice, adversaries do not possess this capability, making it impossible for them to apply our method. Second, even though our work improves our understanding of unintended memorization, we believe that this will benefit privacy researchers more than adversaries. Indeed, it can help practitioners to better audit models under development, and empirical defenses could be derived from our work in the future.

**Limitations and future works** Our method is specifically applicable to classification tasks. Most of our experiments focus on LLMs fine-tuned for textual classification, applied to multiple-choice questions. This setting was chosen because datasets such as MMLU are known to be challenging for language models and are often used to evaluate models' abilities. However, it would be interesting to explore whether our method can be modified to be applicable to LLMs trained on generative tasks. This scenario is indeed widely used and poses significant privacy risks.

Moreover, the approach we developed to predict memorization from the early stages of training could be used to develop empirical defenses. Several methods have already been proposed to mitigate unintended memorization in practice, achieving good privacy-utility trade-offs (Chen et al., 2022; Tang et al., 2022; Chen & Pattabiraman, 2024; Aerni et al., 2024). Our algorithm could be employed to design adaptive defenses that concentrate their efforts on most vulnerable samples to improve the privacy-utility trade-off.

**Reproducibility statement** We have detailed all essential hyperparameters necessary to reproduce our experiments. In addition, we provide the following repository containing the Python, Bash and Slurm scripts that we used to deploy our experiments on an HPC cluster. We also provide a PyPI package containing an automated estimator of PSMI that can be used in a wide range of scenarios.

```
hidden_github_url_experiment_repo_for_review
hidden_github_url_pypi_package_for_review
```

## 5 CONCLUSION

In this work, we demonstrate that it is possible to predict which samples will be memorized by a language model in a classification setting. Our metric is computationally efficient, and it can be utilized from the early stages of training. We provide a theoretical justification for our approach, and we validate its effectiveness on three different language model architectures fine-tuned on three different classification datasets. Moreover, we demonstrate that our method is easily applicable to other classification scenarios by successfully applying it, without modification, to a vision model trained from scratch. We view this method as a first step towards developing useful tools to evaluate models during training, understand the privacy risks they entail, and prevent unintended memorization in the most efficient way.

*Hidden acknowledgements for double-blind reviews.*

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

## A    DEFINING AND MEASURING MEMORIZATION FOR CLASSIFICATION TASKS

Defining and measuring memorization for classification tasks is a challenging task. As explained in Section 1.1, vulnerability to extraction attacks is rarely used for such models. Conversely, in these settings, it is standard to define memorization as vulnerability to membership inference attacks such as LiRA or to use counterfactual memorization. In Section A.1, we present two variants of LiRA (Carlini et al., 2022a): a local version (used in the main body of the paper), which targets a fixed model and its training set, and a global version, which targets a dataset used to train multiple models. In Section A.2, we compare LiRA and counterfactual memorization. It appears that these two definitions are consistent with each others, especially for highly memorized samples. This confirms the relevance of choosing LiRA as the ground truth memorization for our experiments.

### A.1    LIRA ATTACK

#### A.1.1    ATTACKING A MODEL: LOCAL VERSION

In this section we present the original version of LiRA (Carlini et al., 2022a). We call it the *local* version, because it targets a fixed model and tries to determine if a target sample was part of its training set. Note that this setting is aligned with our threat model (See Section 1.2): the model is fixed; and for each sample, if the attack confidently predict that it was part of the training set, we say that it is memorized. This is why we used this *local* version in the main body of this paper.

**Notations**    Let $\mathbf{X} = \{(x_i, y_i)\}_{i \in [\![1,N]\!]}$ be a training set of $N$ labelled elements. We focus on multi-choice question (MCQ) academic benchmarks such as MMLU (Hendrycks et al., 2021b). Let $S$ be a random variable representing a subset of elements in $[\![1, N]\!]$. Let $\mathbf{X}_S = \{(x_i, y_i) \mid i \in S\}$ be the corresponding subset of training elements, and $f_S \sim \mathcal{T}(X_S)$ be a model trained on this subset with the randomized training procedure $\mathcal{T}$. Then, let $\mathcal{L}(x, f_S)$ be the logit gap of the evaluation of $x$ with model $f_S$, i.e. the difference between the highest and second-highest logit.

**The Likelihood Ratio Attack (LiRA)**    Let fix a target subset $S^*$, a target model $f_{S^*} \sim \mathcal{T}(\mathbf{X}_{S^*})$ trained on these elements, and a target element $x \in \mathbf{X}$. As every membership inference attack, LiRA aims to determine whether $x$ was in $\mathbf{X}_{S^*}$. First, we train a great number of *shadow models* $f_S$ on random subsets of $\mathbf{X}$, and evaluates the logit gap $\mathcal{L}(x, f_S)$ for theses shadow models. Then, we gather $\mathcal{L}^{\text{in}} = \{\mathcal{L}(x, f_S) \mid x \in S\}$, the logit gaps of model that were trained on $x$ ; and $\mathcal{L}^{\text{out}}$ for

models that were not trained on $x$. We model these two sets as Gaussian distributions, and compute the probabilities $p^{\text{in}}$ and $p^{\text{out}}$ of the target logit gap $\mathcal{L}(x, f_{S^*})$ under these distributions.

The original LiRA score of Carlini et al. (2022a) is defined as $\text{LiRA}(x, f_S^*) = p^{\text{in}}/p^{\text{out}}$. However, it takes very high and low values positive values. For convenient representations in our graphs, we used the natural logarithm of this score in the main body of this paper. A value greater that $0$ indicates that the sample is memorized, because $p^{\text{in}} > p^{\text{out}}$. For example, a value of $4$ indicates strong memorization, because it means that $p^{\text{in}} \geq e^4 \cdot p^{\text{out}} \simeq 54.6 \cdot p^{\text{out}}$. In other words, the attack suggests that it is $54.6$ times more likely that the target samples belongs to the dataset of the target model, which is significant. This is why, unless otherwise stated, memorized samples are defined as the ones with log-LiRA $\geq 4$ for our experiments in Section 3.

The number of shadow model needed to compute LiRA score is an important hyperparameter. In our experiments, we used 100 shadow models to evaluate memorization in each setting, which is in line with the empirical findings of Carlini et al. (2022a).

### A.1.2 ATTACKING A DATASET: GLOBAL VERSION

It is also possible to use another version of LiRA, as in (Carlini et al., 2022b) for example. We call it the *global* version, because it does not target a fixed model; on the opposite, it attacks multiple models trained on a random splits of the same datasets, and measures the attack success rate of LiRA against each samples, which is defined below.

**The Attack Success Rate (ASR)** It indicates whether a given element $x \in \mathbf{X}$ is likely to be memorized by any model trained on a subset $\mathbf{X}_S$ with training procedure $\mathcal{T}$. Let $\mathcal{D}$ be the distribution of $S$ corresponding the choosing a random subset of $\lfloor N/2 \rfloor$ elements in $[\![1, N]\!]$, meaning that every element is selected with probability 50%. For every target element $x$, the attack success rate is computed as follows:

$$\text{ASR}(x) = \mathbb{P}_{S \sim \mathcal{D}, \; f_S \sim \mathcal{T}(\mathbf{X}_S)} \left[ \mathbb{1}[p^{\text{in}} > p^{\text{out}}] = \mathbb{1}[x \in \mathbf{X}_S] \right] \tag{7}$$

The global LiRA attack represents the likelihood that a sample in a dataset gets memorized by any model trained with a given procedure. As a result, this score is not consistent with our threat model. Indeed, in our threat model we want to audit a *fixed* model, because this is what practitioners do. This is why we did not use the global version in the main body of this paper.

### A.2 COMPARING SEVERAL DEFINITIONS OF MEMORIZATION

We compare two definitions of memorization: counterfactual memorization (Feldman & Zhang, 2020; Zhang et al., 2023) and vulnerability to LiRA membership inference attack (Carlini et al., 2022a). Counterfactual memorization is a *global* measure of memorization. Indeed, it quantifies the impact of a given sample $x$ being in the training set on a population of model trained on random splits of a dataset. Similar to the global variant of LiRA (see Section A.1.2), it is not in line with our threat model, because practitioners want to audit a *fixed* model, and not a population of model trained on random splits of a dataset. Note that because counterfactual memorization is a global definition, we compared it to the global version of LiRA. We recall that this is not the one used in the main body of this paper (see Section A.1). We used Equation 2 in (Zhang et al., 2023) to define counterfactual memorization. We used the logit gap as the performance metric $M$ in their equation.

Our results are presented in Figure 6. We use Spearman's R score to quantify the consistency between the two definitions. Indeed, we are interested in the *order* of samples with respect to the memorization metric. We observe that Spearman's R between the two definition is high in every settings: it is always greater than $0.75$, and escalates to $0.88$ for Mistral models trained on ARC dataset. This demonstrates that LiRA and counterfactual memorization are consistent with each other. In addition, we can separate the samples in two groups: a first, weakly memorized group, for which there is greater variability between the two definitions (bottom left of the graphs), and a strongly memorized group, for which the two definitions are much more consistent with each other (top right of the graphs). The second group is the most important one in our setting, because we are interested in predicting memorized samples (and not predicting non-memorized ones).

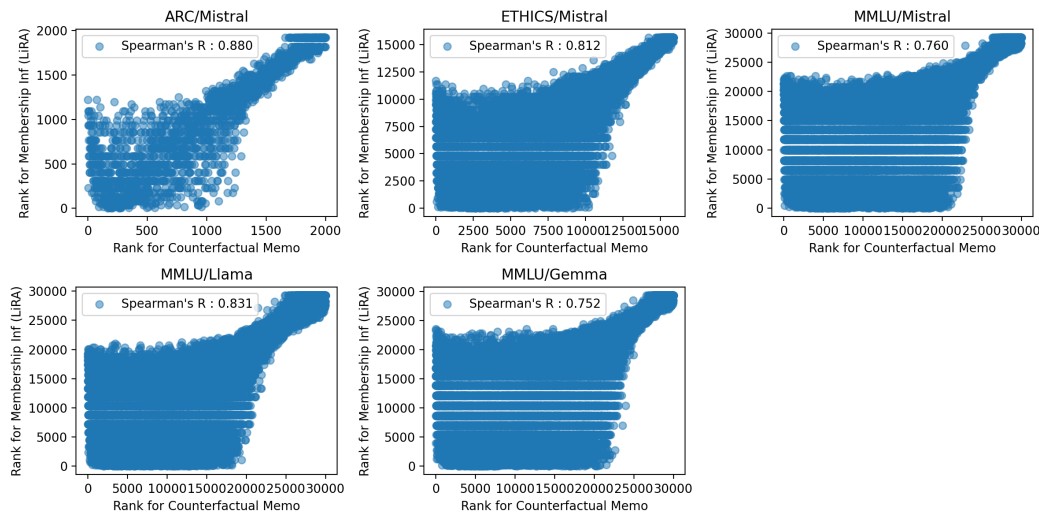

Figure 6: Comparing two definitions of memorization: counterfactual memorization (Feldman & Zhang, 2020; Zhang et al., 2023) and LiRA membership inference (Carlini et al., 2022a). We measure Spearmans'R coefficient to evaluate the consistency between the definitions. These experiments are conducted on models trained for 10 epochs.

The coherence of these two definitions, especially for highly memorized samples, confirms the relevance of choosing LiRA as the ground truth memorization for our experiments.

## B    PROOF OF THEOREM 1 AND DISCUSSION

In this section we prove Theorem 1 and discuss it. We recall the theorem:

**Theorem 1** *Let $(X, Y) \in \mathbb{R}^d \times \{0,1\}$ be random variables. We assume that $p(Y = 0) = p(Y = 1) = 0.5$ and that $X$ is a continuous random variable. We also assume that there exist $\mu_0, \mu_1 \in \mathbb{R}^d$ with $\mu_0 \neq \mu_1$, and $\Sigma_0, \Sigma_1 \in \mathbb{R}^{d \times d}$, and a Bernoulli variable $\Delta \sim \mathcal{B}(\varepsilon)$ with $\varepsilon \in ]0, 1[$ such that:*

$$p(X \mid Y = 0, \Delta = 0) \sim \mathcal{N}(\mu_0, \Sigma_0) \quad \text{and} \quad p(X \mid Y = 1, \Delta = 0) \sim \mathcal{N}(\mu_1, \Sigma_1) \tag{3}$$

$$\forall x \in \mathbb{R}^d, \quad p(Y = 0 \mid \Delta = 1, X = x) = p(Y = 1 \mid \Delta = 1, X = x) = 0.5 \tag{4}$$

*Given this, we then have:*

$$\mathop{\mathbb{E}}_{X,Y} \left[ \text{PSMI}(X, Y) \mid \Delta = 0 \right] > 0 \tag{5}$$

$$\mathop{\mathbb{E}}_{X,Y} \left[ \text{PSMI}(X, Y) \mid \Delta = 1 \right] = 0 \tag{6}$$

**Proof of Equation 6**    Let $x, y \in \mathbb{R}^d \times \{0, 1\}$. We use the hypothesis we made in Equation 4:

$$p(X = x, Y = y \mid \Delta = 1) = p(Y = y \mid \Delta = 1, X = x) \times p(X = x \mid \Delta = 1) \tag{8}$$
$$= 0.5 \times p(X = x \mid \Delta = 1) \tag{9}$$
$$= p(Y = y \mid \Delta = 1) \times p(X = x \mid \Delta = 1) \tag{10}$$
$$\tag{11}$$

Consequently, given $\Delta = 1$, $X$ and $Y$ are independent. We conclude that the expected value of PSMI is zero, which proves Equation 6.

$$\underset{X,Y}{\mathbb{E}}[\text{PSMI}(X,Y) \mid \Delta = 1] = \int\limits_{X,Y} \int\limits_{\theta \sim \mathcal{U}(\mathbb{S}^d)} \log \frac{p(\theta^T x, y)}{p(\theta^T x)p(y)} \mathrm{d}p(X,Y \mid \Delta = 1)\mathrm{d}p(\theta) \qquad (12)$$

$$= \int\limits_{X,Y} \int\limits_{\theta \sim \mathcal{U}(\mathbb{S}^d)} \log \frac{p(\theta^T x)p(y)}{p(\theta^T x)p(y)} \mathrm{d}p(X,Y \mid \Delta = 1)\mathrm{d}p(\theta) \qquad (13)$$

$$= 0 \qquad (14)$$

**Proof of Equation 5** First, we have:

$$\text{SMI}(X,Y) = \mathbb{E}[\text{PSMI}(X,Y)] \qquad (15)$$
$$= \mathbb{E}[\text{PSMI}(X,Y) \mid \Delta = 0]p(\Delta = 0) + \mathbb{E}[\text{PSMI}(X,Y) \mid \Delta = 1]p(\Delta = 1) \qquad (16)$$

Using Equation 6 that we have proved, we obtain:

$$\mathbb{E}[\text{PSMI}(X,Y) \mid \Delta = 0] > \text{SMI}(X,Y) \qquad (17)$$

As a result, it is sufficient to demonstrate Equation 5 with $\text{SMI}(X,Y)$ instead of $\mathbb{E}[\text{PSMI}(X,Y) \mid \Delta = 0]$. To do this, we will apply Theorem 1 in (Wongso et al., 2023b). To do this, we search $(R_0, R_1, m_g, \nu) \in \mathbb{R}_{+,*}^4$ such that $(X,Y)$ is $(R_1, R_2, m_g, \nu)$-SSM separated with respect to Definition 3 in (Wongso et al., 2023b). Let $D = ||\mu_0 - \mu_1||$. Using $\mu_0$ and $\mu_1$ and the centers of the spheres, this means that $(R_0, R_1, m_g, \nu)$ should satisfy:

$$p(||X - \mu_0|| > R_0) = p(||X + \mu_1|| > R_1) = \nu \quad \text{and} \quad R_0 + R_1 + m_g = D \qquad (18)$$

There are many values of $(R_0, R_1, m_g, \nu)$ which satisfy these conditions. When applying Theorem 1 in (Wongso et al., 2023b), these values give different lower bounds. Here is an algorithm to create a valid tuple $(R_0, R_1, m_g, \nu)$ given a hyperparameter $R \in ]0, D/2[$.

1. Let $S_0$ (resp. $S_1$) be the sphere of center $\mu_0$ (resp. $\mu_1$) and radius $R$.

2. Let $\nu_0 = p(X \in S_0 \mid Y = 0)$ and $\nu_1 = p(X \in S_1 \mid Y = 1)$. Given the Gaussian assumptions we made in Equation 3, we have $\nu_0, \nu_1 \in ]0, 1[$.

3. Let $i \in \{0, 1\}$ and $j = i - 1$ such that $\nu_i \geq \nu_j$. We fix $R_i = R$ and $\nu = \nu_i$.

4. We will now start with $R_j = R$ and decrease its value until Equation 18 is satisfied. Because $X$ is a continuous random variable, the following function is continuous, decreasing, equal to 1 when $t = 0$, and because $\nu_j \leq \nu_i$, its value is $\leq \nu$ for $t = 1$:

$$t \in [0, 1] \mapsto p(||X - \mu_j|| > t \cdot R \mid Y = j) \qquad (19)$$

5. As a consequence, due to the intermediate values theorem, there exists $t_j$ in $]0, 1]$ such that $p(||X - \mu_j|| > t \cdot R \mid Y = j) = \nu$.

6. We set $R_j = t \cdot R$ and $m_g = D - R_0 - R_1$. Because $R_0, R_1 \leq R < D/2$, we have $m_g > 0$

7. Now, we can apply Theorem 1 in (Wongso et al., 2023b) :

$$\text{SMI}(X,Y) > (1 - H(\nu, 1 - \nu)) \times B_{\gamma(m_g, R_0, R_1)}\left(\frac{d-1}{2}, \frac{1}{2}\right) \qquad (20)$$

Where:

- $H$ is the entropy function $H(p_1, p_2) = -p_1 \log p1 - p_2 \log p_2$. We can easily prove that $(1 - H(\nu, 1 - \nu)$ is convex on $]0, 1[$ and that its minimal value is $> 0$.

- $\gamma(m_g, R_0, R_1) = \frac{m_g}{m_g + R_0 + R_1}\left(2 - \frac{m_g}{m_g + R_0 + R_1}\right) = \frac{m_g}{D}\left(2 - \frac{m_g}{D}\right) \in ]0, 1[$

- $B$ is the incomplete beta function defined as follows. Because $\gamma(m_g, R_0, R_1) \in ]0, 1[$, it is clear that its value is always $> 0$.

$$B_\gamma(a, b) = \int_0^\gamma t^{a-1}(1-t)^{b-1}\mathrm{d}t \tag{21}$$

This proves that $\mathrm{SMI}(X, Y) > 0$, which demonstrates Equation 6 and concludes the proof.

$\square$

**Discussion on a better bound for Equation 5** The proof above provides a constructive algorithm to obtain $(R_0, R_1, m_g, \nu) \in \mathbb{R}^4_{+,*}$ such that $(X, Y)$ is $(R_1, R_2, m_g, \nu)$-SSM separated with respect to Definition 3 in (Wongso et al., 2023b). Depending on the hyperparameter $R \in ]0, D/2[$, the bound is different. As a result, this hyperparameter can be optimized to find the better possible bound with this algorithm. We did not performed this optimization because it is not useful for the purpose of Theorem 1. Indeed, we only use this theorem to illustrate why we expect outliers in the hidden representations distribution to have a lower PSMI (see Section 2.1).

## C  IMPLEMENTATION DETAILS

To help reproducing our results, we provide a GitHub repository containing the Python source code of our experiments, as well as the Bash and Slurm scripts to deploy them on a HPC cluster.[2] We also provide a PyPI package containing an automated estimator of PSMI that can be used in a wide range of scenarios.[3]

In this section we discuss how we implemented our experiments in practice. In Section C.1, we discuss how we adapted the baseline of Biderman et al. (2023) to classification, in Section C.2, we discuss how we implemented our measures of memorization, in Section C.3 we elaborate on the implementation of our predictors.

### C.1  IMPLEMENTING OUR BASELINE

As explained in the introduction, the baseline of Biderman et al. (2023) is the only comparable method we are aware of. However, it is not directly applicable to our classification setting. Their method measures $k$-extractability (Carlini et al., 2021) on the partially trained model to predict memorization in the fully trained model. However, as explained in Section 1.1, extractability is rarely used to define memorization in a classification setting. Indeed, current extraction or reconstruction attacks against classification models are both more complex and less powerful than extraction attacks against generative models (Carlini et al., 2023). Consequently, we modified the baseline of Biderman et al. (2023) to suit our classification setting. While we still use memorization within the partially trained model to predict memorization in the final model, we replaced $k$-extractability by the vulnerability to LiRA attack.

The computational cost of this adapted baseline is significantly higher than that of the methods we evaluate, as it requires training the shadow models needed for LiRA attack. As a consequence, this baseline would not be suitable for practitioner within our threat model (see Section 1.2). Nevertheless, we compare our method to this baseline because it is the only comparable approach that assess the possibility of predicting memorization before the end of training.

### C.2  IMPLEMENTING MEMORIZATION MEASURES

The local version of LiRA, the global version, and counterfactual memorization all require a large number of shadow models (see details in Section A). To minimize the computational cost of our experiments, for each dataset, we trained 100 shadow models on random splits containing half of the elements of the dataset. Each random split (and the model trained on it) is associated to a number

---

[2]`hidden_github_url_experiment_repo_for_review`
[3]`hidden_github_url_pypi_package_for_review`

between 0 and 99 (see `split_id` attribute in `training_cfg.py`) corresponding to the seed of the random split.

- **Local LiRA:** We select the model trained on random split $0$ to be our target model, and use the 99 other models as the shadow models for the attack. In addition to the training cost, this requires one forward pass per shadow model on the training set of the target model (i.e. random split $0$). Note that this is the setting used in the main body of the paper, so the PSMI is computed on this random split $0$ and used to predict memorization for the model trained on it.

- **Global LiRA:** We attack each model with the 99 other models trained on different random splits, and measure the attack success rate on each sample $x$ of the dataset. In addition to the training cost, this requires one forward pass per shadow model on *all* elements of the dataset.

- **Counterfactual memorization:** For each element $x$ of the complete dataset, we separate the shadow models into two groups: the one that had $x$ in the training set, and the others. Given that each random split contains half of the samples, these two groups have roughly the same size. We used them to compute counterfactual memorization (Zhang et al., 2023). In addition to the training cost, this requires one forward pass per shadow model on *all* elements of the dataset.

Note that in our GitHub repository, we use the term `dynamic` to describe local measures such as the local version of LiRA or the PSMI of the target model; and we use the term `static` to describe global measures on a population of models such as the global version of LiRA or counterfactual memorization. The training of these shadow model was by far the most expensive part of our experiments from a computational perspective. However, this operation can be parallelized on a many workers within an HPC cluster, because each shadow model is trained independently.

## C.3 IMPLEMENTING PREDICTORS

Algorithm 1 in Section 2.3 explains how we use PSMI to predict memorization in the final model. This algorithm can be easily adapted to use other predictors instead of PSMI. For instance, Algorithm 2 illustrates how to use the loss as a predictor. Unlike PSMI, it does not require a hyperparameter to select a layer. However, as shown in Section 3.2, the choice of the last layer is robust across all empirical settings we evaluated, so this hyperparameter does not introduce additional complexity. Conversely, using the loss instead of PSMI requires an extra hyperparameter (denoted $x$ in Algorithm 2) to define the proportion of samples to filter based on their loss values.

---

**Algorithm 2** Using Loss to predict memorization

---

1: Interrupt training when the median training loss has decreased by at least **95%**.
2: Compute a forward pass for every sample to retrieve the **loss**.
3: Predict that every sample with **top $x$%** highest loss will be memorized.

---

We evaluated five possible metrics to predict memorization at the early stages of training. For the metrics that require the hidden states of the model (PSMI and Mahalanobis distance), we recall that the hidden state at layer $k$ is defined as the representation of the last token (the one before the label) after layer $k$ (see section 2.1).

- **PSMI.** We use algorithm 1 in (Wongso et al., 2023a) to estimate PSMI. We sample 2000 direction uniformly on the unit sphere. Indeed, we observed that the mutual information between random directions and the label has a mean of about $4.5 \cdot 10^{-3}$ and a standard deviation of about $5.5 \cdot 10^{-3}$. Thus, if we approximate these distributions by Gaussians, we get a margin at 95% confidence interval of about $(1.96 \times 5.5 \cdot 10^{-3})/\sqrt{2000} \simeq 2.4 \cdot 10^{-4}$. This is about 20 times smaller than the mean, so we consider that our metric is stable enough with 2000 estimators.

- **Loss.** We directly use the cross-entropy loss of the model for the last token before the label. This metric was suggested by Leemann et al. (2024).

- **Logit Gap.** The logits are the outputs of the fully-connected layer applied to the last token, before the softmax. We define the logit gap as the difference between the logit of the correct prediction and the maximum logit of an incorrect prediction.

- **Mahalanobis distance.** It is the Mahalanobis distance (Mahalanobis, 1936) of the hidden representation of a training sample to the distribution of hidden representation of the other training samples. To reduce computational costs, we first project every hidden states using a Principal Component Analysis (PCA) with a target dimension of 500. This metric was suggested by Azize & Basu (2024).

- **Our baseline: early memorization.** We define early memorization as the natural logarithm of LiRA attack against the partially trained model. See Appendix C.1.

## D  ADDITIONAL EXPERIMENTS

### D.1  TPR/FPR TRADE-OFFS FOR EACH SETTING AT EVERY CHECKPOINT

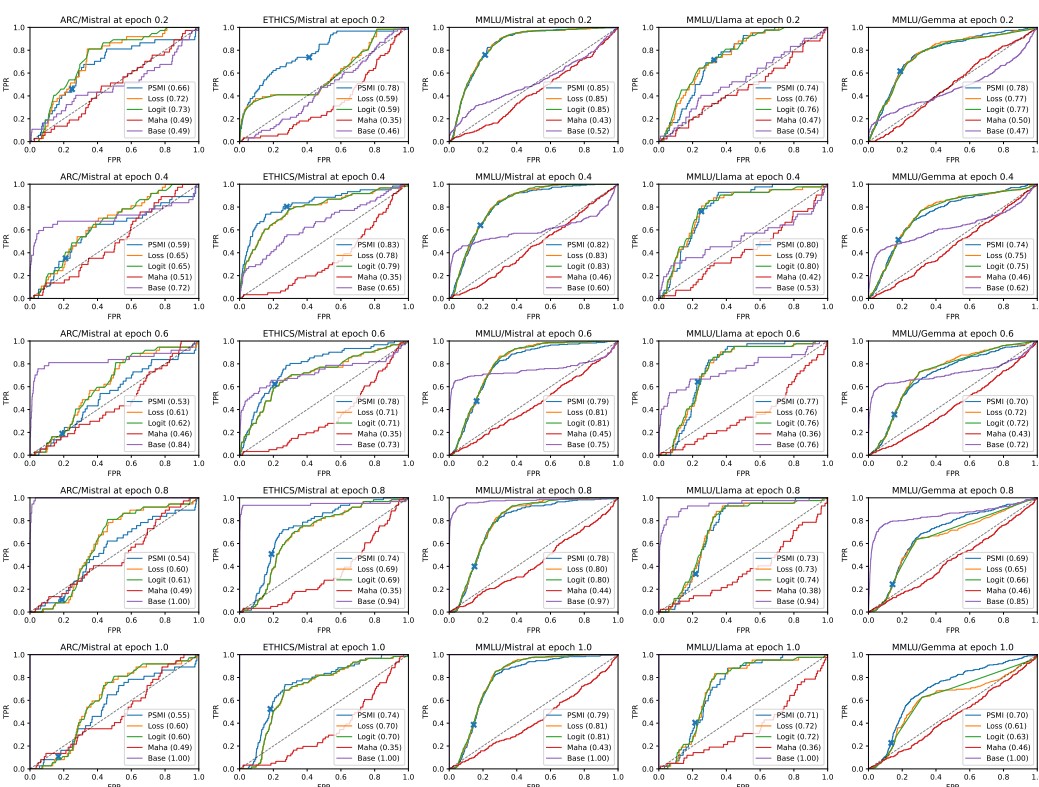

Figure 7: TPR/FPR trade-offs for each setting at every checkpoint. The number in parentheses corresponds to the AUC of the curve. The blue cross indicates the result using the default hyperparameters of Algorithm 1. The AUC of the baseline ("Base") converges to 1.0 at epoch 1.0 because, at that stage, the baseline is the same as what we are trying to predict. We remind that practitioners within our threat model do not have the resources to compute the baseline and instead attempt to approximate it using other metrics early in the training pipeline.

### D.2  ADDITIONAL RESULTS ON THE DYNAMICS OF TRAINING

We always interrupt training when the median training loss has decreased by 95%, and measure ground truth memorization after 1 epoch of training (see Section 3). To validate this choice, we conducted the experiments described in Section 3.2. Figure 8 presents additional plots for experimental settings not discussed in that section. These results confirm that memorization can be predicted early in the training pipeline and that memorized samples have not yet been memorized at that point.

|  | Epoch 0 | Epoch 0.2 | Epoch 0.4 | Epoch 0.6 | Epoch 0.8 | Epoch 1 |
|---|---|---|---|---|---|---|
| ARC/Mistral | 0.000% | 93.724% | 98.015% | 99.397% | 98.907% | 99.222% |
| ETHICS/Mistral | 0.000% | 86.036% | 95.198% | 98.985% | 99.665% | 99.739% |
| MMLU/Mistral | 0.000% | 98.967% | 99.776% | 99.916% | 99.916% | 99.895% |
| MMLU/Llama | 0.000% | 91.674% | 98.186% | 99.329% | 99.336% | 99.267% |
| MMLU/Gemma | 0.000% | 99.543% | 99.606% | 99.855% | 99.980% | 99.979% |

Table 1: Decrease in the median training loss relative to epoch 0 throughout training.

**Special case of ARC/Mistral**   Table 1 presents the decrease of the median training loss relative to epoch 0 throughout training. We saved models every 0.2 epoch to analyze them and measure their performance. We observe that for ARC/Mistral, the median training loss has decreased by 93.724% at epoch 0.2, which is close to 95%. Conversely, by epoch 0.4, the median training loss has decreased by significantly more than 95%. This is why, for that setting, we predict memorization at epoch 0.2, the checkpoint where the decrease is closest to 95%. For the other settings, as indicated in Section 3, we predict memorization at the first checkpoint where the median training loss has decreased by at least 95%.

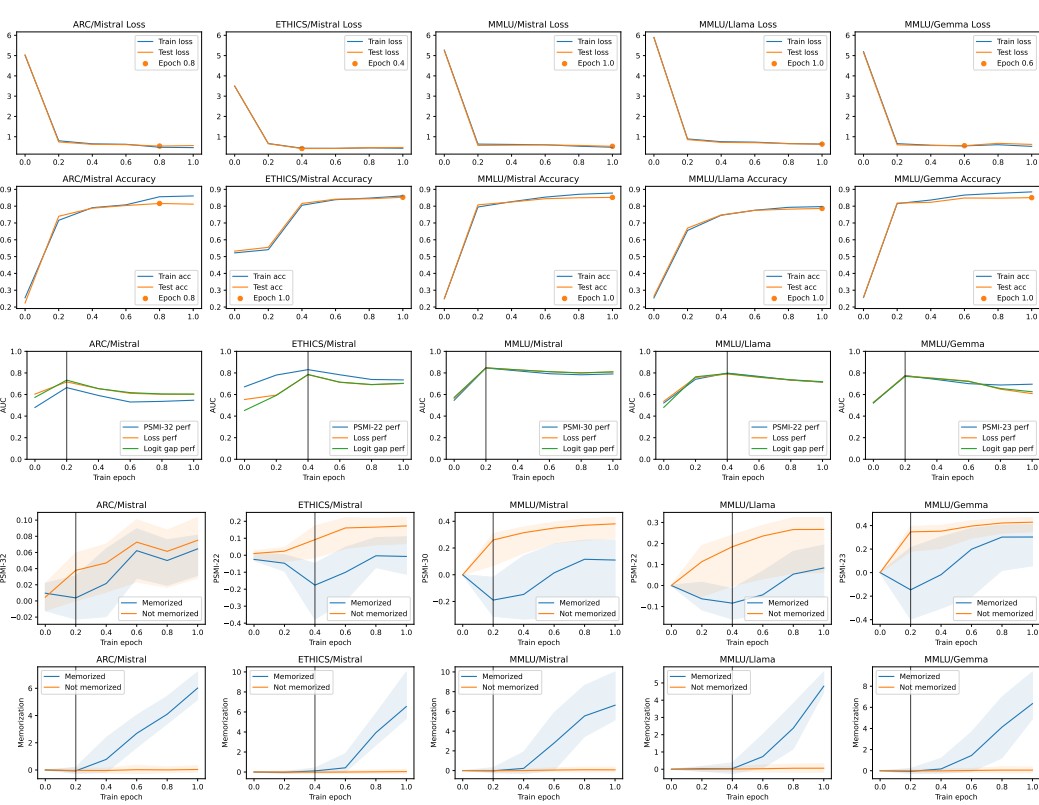

Figure 8: Additional results related to the dynamics of training and the appropriate moment to interrupt training. **First row:** Training loss, testing loss, and epoch of the best testing loss for each experimental setting. **Second row:** Training accuracy, testing accuracy, and epoch of the best testing accuracy for each experimental setting. **Third row:** AUC of PSMI, Loss and Logit Gap for predicting memorization. The vertical line indicates the point at which training loss has decreased by 95%, marking the moment when training is stopped to predict memorization. **Fourth row:** The solid line shows the median PSMI for samples that will be memorized or not within the fully trained model. The shaded area represents the 25%-75% quantiles. PSMI is measured at the layer that obtained the highest AUC. **Fifth row:** Similar representation for the memorization within the partially trained model.

## D.3 HISTOGRAMS OF MEMORIZATION THROUGHOUT TRAINING

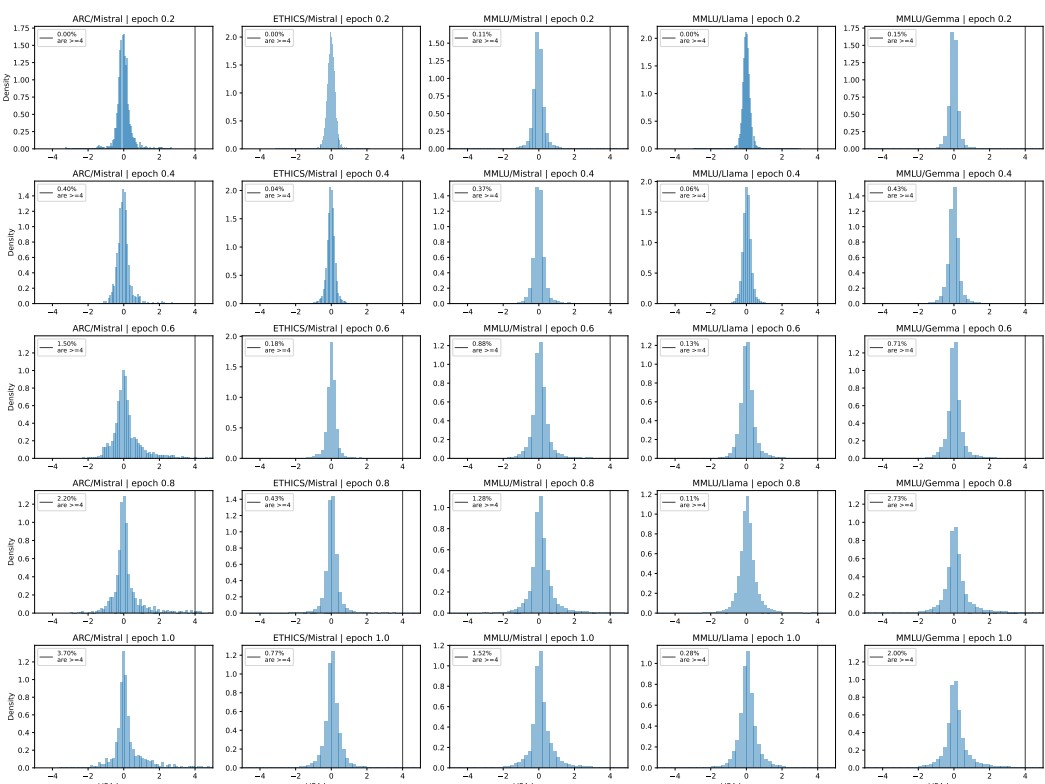

Figure 9: Histograms of memorization throughout training. The legend displays the proportion of samples with log-LiRA $\geq 4$, which is the threshold used to define memorization in all figures unless otherwise specified.

## D.4 ADDITIONAL RESULTS ON THE IMPACT OF THE MEMORIZATION THRESHOLD

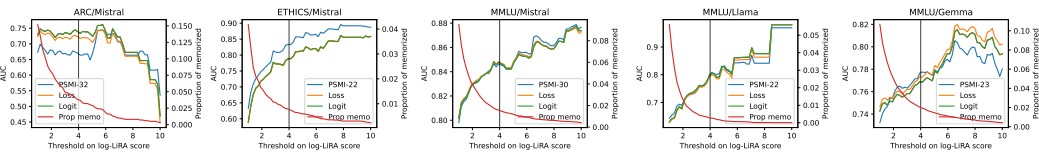

Figure 10: Impact of the threshold used to define "memorized" and "non memorized" samples. The vertical bar indicates the default threshold log-LiRA = 4.

## D.5 ABLATION STUDY ON THE LAYERS FOR MAHALANOBIS DISTANCE

Similar to the approach in Section 3.2 and Figure 4b, we conducted an ablation study on the layers for Mahalanobis distance (see Figure 11). These results were used in the other figures to ensure that the Mahalanobis distance is computed at the layer that maximizes the resulting AUC value.

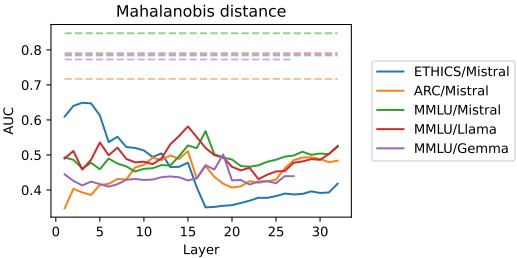

Figure 11: Impact of the choice of layer on the AUC using the Mahalanobis distance. The dashed lines represent the AUC with the loss, which is independent of the layer.

### D.6 ADDITIONAL RESULTS WITH CIFAR-10

Figure 12 presents additional results obtained by applying our method as-is to a wide residual network trained from scratch on CIFAR-10. We vary the threshold applied to log-LiRA to define memorized samples. We observe that our method becomes increasingly effective as the samples to be detected become more highly memorized. It converges towards the experiment on the right of the figure, where memorized samples are defined as the canaries crafted by (Aerni et al., 2024) to mimic the most vulnerable samples.

For these experiments, the model was trained for 300 epochs, and we interrupted training after 4 epochs, when the median training loss had decreased by 95%.

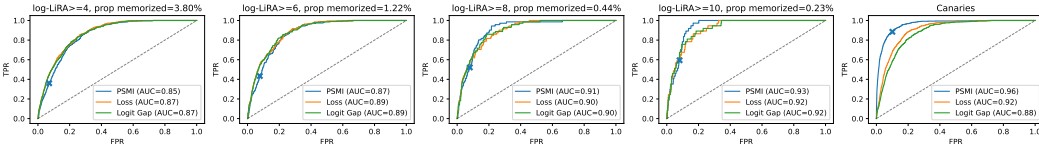

Figure 12: Predicting memorized samples on a WRN16-4 (Zagoruyko & Komodakis, 2016) trained from scratch on CIFAR-10 using the framework of Aerni et al. (2024). In the first four graphs, we vary the threshold applied to log-LiRA to define memorized samples. The title displays the proportion of memorized samples in the fully trained model using this definition. The blue cross marks the performance of the default hyperparameters from Algorithm 1. The last graph presents the same experiment, where memorized samples are defined as the canaries inserted by Aerni et al. (2024) to mimic the most vulnerable samples in the training set.

