# OpenReview forum: "Predicting and analyzing memorization within fine-tuned Large Language Models"
_ICLR.cc/2025/Conference — Submitted to ICLR 2025_

### Official Review · Reviewer_ZnJf · 2024-10-26

**Soundness:** 3
**Presentation:** 4
**Contribution:** 1
**Rating:** 3
**Confidence:** 3

**Summary:**

Given a fine-tuning dataset D and a language model M which is part-way through being fine-tuned on D, this work introduces an information-theoretical metric for classifying which samples in D have been memorized by M. This technique is evaluated based on its FPR at TPR=0.75, and, according to this evaluation, is shown to outperform baseline techniques for detecting memorized samples. The authors also include an analysis memorization dynamics throughout training and an ablation study.

**Strengths:**

1. This paper is extraordinarily well-written. The key points, relation to prior work, and implementation details were all very clearly explained.
2. I greatly appreciated the ablation study, which effectively justified the threshold for stopping training once loss has dropped by 95%. It also clarified the effect of the layer at which activations are extracted.
3. The paper is thorough in general, with comparisons against multiple baselines and many informative appendices.
4. The authors clearly explain the problem they are trying to solve: providing a compute-efficient classifier for memorized samples which has low FPR at a fixed high TPR.

**Weaknesses:**

1. The PSMI metric proposed here does not improve very much over much simpler baselines. The "loss" and "logit gap" baselines are much simpler (essentially, looking at the training samples on which the model is most performant), but have only a very slightly higher FPR than PSMI at the chosen TPR=0.75.
2. Moreover, this effect appears very sensitive to the fixed choice of TPR. Typically when I've seen other authors choose a "fixed high TPR" they select something like 99%; but at TPR=0.99 it appears that the PSMI metric introduced here would perform substantially worse than the "loss" and "logit gap" baselines. At intermediate TPRs between 0.75 and 1, it doesn't seem that there's a clear best method between PSMI, loss, and logit gap. In fact, it seems that TPR=0.75 is approximately the TPR at which PSMI looks best relative to the baselines.
3. The Mahalanobis distance baseline uses activations from layer 13. On the other hand, the performant methods all use activations from the final layers of the model (including its output). Based on the ablation study, it seems likely that the only effect in the Mahalanobis vs. PSMI comparison is the choice of layer. I request that the authors rerun the Mahalanobis distance baseline using activations from the same layer as PSMI.
4. The authors argue that PSMI is a good choice of metric because it comes with a distinguished theoretically-justified threshold. While this is true, I am unconvinced that it is important. In practice, practitioners can choose the amount of data that they would like to filter out (e.g. perhaps they are willing to filter out 1% of their data due to privacy concerns), and they can select their threshold in order to filter out that amount of data. Between this and my point (2), I don't see practical advantages for PSMI.
5. Given that the "hard" samples in section 3.1 are *defined as* being those for which PSMI is small at the training cut-off, I find the analysis based on later changes in PSMI (e.g. seeing that it increases) unconvincing.

**Questions:**

1. It seems to me that equation (6) in Theorem 1 could be tightened to an equality rather than an inequality. Here is my argument: after swapping the orders of the expected values, the LHS of (6) is just the mutual information between X and Y conditional on $\Delta=1$. But by assumption, these variables are independent when $\Delta=1$, so the mutual information should be 0. If this is incorrect, can the authors provide a counterexample?
2. Given that PSMI is estimated by fitting Gaussians to $p(\theta^Tx_k|y)$, it should be closely related to the Mahalanobis distance between $x_k$ and $X_k|Y$. Could the authors explain this relationship? I would find it informative for getting intuition about PSMI.

---

> ### Author Response · Authors · 2024-11-23
> **Answer to ZnJf**
>
> Dear Reviewer,
>
> Thank you for your valuable feedback. We have provided a global response to the four reviews we received. This message offers additional details addressing the specific concerns you raised.
>
> Regarding the weaknesses you have highlighted:
> 1. Indeed, PSMI does not improve significantly over the loss and the logit gap. These three metrics achieve similar performance and significantly outperform our baseline adapted from Biderman et al., which is the only comparable approach we are aware of. We have changed the narrative to communicate that all three methods can be used (see global answer).
> 2. You are absolutely right, the choice of TPR=75% is arbitrary. We did not choose TPR=99% because predicting memorization is a challenging task, and it would result in a very high FPR, which our practitioner cannot afford. As discussed in our global answer, we switched to the AUC, removing the dependency on the threshold set on TPR.
> 3. Layer 13 corresponded to the one for which the Mahalanobis distance was the most performant in that setting (see line 338. Now, it is layers 23 and 19 because we are running experiments with only one epoch of training). We conducted an ablation study on the layers for the Mahalanobis distance in Appendix D2.
> 4. You are correct that there are many scenarios where the loss would be easier to use than PSMI. We have changed the narrative to communicate that all three methods can be used and that they outperform our baseline.
> 5. Section 3.1 indeed had weaknesses. We have removed it and replaced it with more factual discussions (see global answer). We are now considering "memorized" and "non-memorized" samples, which removes the bias you pointed out.
>
> Regarding your questions:
> 1. You are absolutely right. We have improved the theorem thanks to your remark.
> 2. For our application scenario, there are two main differences between PSMI and Mahalanobis distance:
>    1. **Computational cost.** PSMI is estimated from Gaussian estimation of 1D projections of the features, which is very efficient. On the other hand, Mahalanobis distance requires estimating a covariance matrix in high dimensions, which is extremely costly. This is why we had to perform a PCA before computing the Mahalanobis distance (see lines 183–187).
>    2. **Incorporation of label information.** PSMI involves comparing $p(\theta^Tx|y)$ and $p(\theta^Tx)$. As a result, for a given label $y$, PSMI considers not only the conditional probability distribution $p(\theta^Tx|y)$ but also the probability distribution for the other labels. In contrast, Mahalanobis distance only measures the similarity between $x$ and $X|Y$, ignoring the probability distribution of $X$ for the other labels. Consequently, PSMI accounts for more information, which explains its better performance.
>
> Please let us know if our revision correctly addresses your concerns. We would be happy to discuss any aspect further.

---

> ### Author Response · Authors · 2024-11-29
> **Further discussion would be greatly appreciated**
>
> Dear Reviewer ZnJf,
>
> Thank you very much for the time and effort you have dedicated to reviewing our paper. We greatly appreciate your constructive feedback. Given the extended discussion period, we believe that further correspondence with you would add significant value to our manuscript.
>
> We hope that our rebuttal has sufficiently addressed the concerns you raised. Please let us know if any questions or issues remain, as we would be more than happy to discuss them further.
>
> Best regards,
>
> The Authors

---

### Official Review · Reviewer_jqv5 · 2024-10-27

**Soundness:** 3
**Presentation:** 2
**Contribution:** 3
**Rating:** 5
**Confidence:** 4

**Summary:**

The authors study how to detect memorization by language models early in finetuning on classification tasks. The authors argue that PSMI is a predictive measure that can be computed early in training.

**Strengths:**

- Grammatical errors aside, the paper is relatively well written
- The Related Work is extensive

**Edit: The authors improved the paper in many of the ways I requested. I'm increasing my score from 3 to 5. I had initially planned to go higher but the newest version of the manuscript helped me realize that this paper is about detecting memorization by classifiers. The focus on LLMs in the title, abstract, introduction, methodology and results is (in my new understanding) a smokescreen/misdirection because the paper applies generically to classifiers, as their new image classifier results make clear.**

**Weaknesses:**

- This work, as best as I can tell, studies memorization in a classification setting. This limits the contribution of the work. It also oddly contrasts with the literature since, to the best of my knowledge, much work on memorization by LLMs has not studied classification and instead studies a broader notation of memorization of sequences of tokens (e.g., Social Security Numbers, credit card information)
- The paper’s narrow focus on classification is not made clear from the title nor from the schematic Figure 1. The title could perhaps be edited to make this clear, or the schematic adapted with a more explicit caption.
- While the Related Work is extensive and the authors should be commended, I couldn’t determine what other works in the memorization literature focus specifically on classification. I feel like this should be made clear to the reader to know which other memorization-detection methods are relevant in the classification setting.
- Why do the authors choose LiRA membership inference attack (an MIA-based approach) as a ground truth for memorization, rather than a memorization based metric? The most famous is probably kl-memorization, used by Biderman et al. 2023. Another might be kl-edit distance, introduced by Duan et al. 2024 (https://arxiv.org/abs/2406.14549) as a softer notion of memorization
- The authors comment that they finetune for 10 epochs (Line 284 to 285). This is a large number of epochs. Many post-training recipes prescribe < 1 epoch. This is further backed up by the author’s Figure 2c, which shows the test loss begins increasing after the first epoch. If, for instance, a practitioner uses a validation split, then they would stop at or before 1 training epoch. Consequently, I feel like the memorization this paper is studying is even more distantly connected from practice. The authors state that the validation accuracy peaks at 5 epochs (“The second [line] (epoch 5) represents the best validation accuracy” on Lines 335-336)  but I don’t see evidence of this. Figure 7 in the Appendix shows that the test loss diverges after Epoch 1 in all settings and the test accuracy plateaus after 1 epoch.
- Section 3.1 feels handwavy and unscientific. Certain sentences seem like word soup. For instance, “The superposition of all patterns learnt by the model results in the decision boundary that separates samples that are classified in each class”; what does superposition mean in this context? This feels like a vacuous statement for saying “the statistical relationship for performing classification results in a decision boundary that separates samples into their classes,” which is trite. The next sentence “The model learns simple patterns, which are efficient to perform the task approximately.” is equally perplexing. What does simple mean? What does it mean for a pattern to be efficient at approximately performing a task? This section needs to be made more quantitative and more precise.
- Section 3.1 also suggests an alternative early measure of detecting memorization: Figure 3.1c suggests that the loss per sample might be as predictive as PSMI. To explain why this might make sense, some data might be “easy” to learn and other data might be “hard” to learn. The hard to learn data are identifiable because (a)  their losses are higher and because (b) their losses exhibit non-monotonic learning curves, notably increasing before more slowly decreasing. If such hard to learn data are more likely to be memorized (as suggested by Figure 3.1a), then this test would identify the same data but be faster than extracting hidden activations of the networks and then bombarding them with vectors sampled from the hypersphere. I normally try to avoid suggesting experiments for authors because experiments can be slow and difficult, but in this case, it appears that the authors already have the loss data; the question is how well the loss performs at predicting memorization (especially compared to PSMI).
- After I finished writing the above paragraph, I made it to Section 3.2 and discovered in Figure 3a that the loss and the logit gap are as predictive of memorization (in this setting) as PSMI. I’m now very perplexed why Section 2 pays so much attention to PSMI when the logit gap and the loss are equally performant, much easier to implement and probably much more computationally efficient. The authors might argue that if we focus on FPR @ TPR = 0.75, PSMI is slightly preferable to Loss (Figure 3b), but (1) the difference is very minor and (2) if we consider a slightly different TPR (e.g. TPR = 0.8), the difference between PSMI, Loss and Logit Gap becomes ~0 and (3) the claim that PSMI is better than loss under FPR @ TPR = 0.75 appears to be refuted by Figure 4a, which shows that FPR @ TPR = 0.75 is lowest for loss on ETHICS/MISTRAL. Edit: Figure 11 in the Appendix shows that PSMI and and Loss are almost indistinguishable in all cases; I’m not sure why the logit gap is omitted.
- I find the argument in “Why PSMI is the most practical estimator” to be unpersuasive because (1) tuning hyperparameters is what we do all the time and the same theoretical justification used for PSMI can probably also be extended to be used for the loss and for the logit gap, (2) because PSMI requires testing which layer(s) can be used, but the loss and logit gap do not, (3) because loss and logit gap don’t require Step 3 of Algorithm 1, making them faster, more memory efficient and with less room for implementation error, and (4) because loss, if it can be used to detect memorization, might be extended beyond the classification setting, whereas PSMI is much hard to extend to a general setting.
- Figure 4b is uninformative. Most of the figure is whitespace.
- The choice of Early Memorization from Biderman 2023 as a baseline is questionable to me. As I understand, that paper uses a different definition of memorization (kl-memorization), is not limited to classification, and is designed for pretraining not finetuning. It’s possible that there is no appropriate baseline for classification. If so, I think it would be good to explicitly state this and say “We’re using this baseline - even though it wasn’t designed for our setting - simply because there are no other baselines.” If I’m mistaken, please correct me.

**I would significantly increase my score if the authors makes the following changes**:

- More prominently communicate that they specifically consider the narrow setting of classification
- Switch to memorization-focused metrics (e.g., KL-memorization)  rather than membership inference attack-focused metrics (e.g., LiRA), or change the narrative
- Stop finetuning for 10 epochs and instead finetune for 1 epoch max, following standard practice and their own appendix results
- Change the narrative to communicate that all three metrics - Loss, Logit Gap, PSMI - can be used to detect memorization
- Empirically study when the three metrics are useful and what their tradeoffs are
- Extend the theoretical analysis to include the loss and logit gap. I believe this should be doable in the authors' toy mathematical model.

**Alternatively, the authors are very welcome to explain to me why my criticisms are incorrect.** The choice is to either change the paper or change my understanding of it :)

The following points are minor suggestions, nits or errors that can hopefully be changed, cleaned or fixed, if the authors agree:
- Line 26 & 33: I believe Large Language Models should be abbreviated LLMs, not LLM.
- Line 29: “articles” is an unusual term for scientific research and more typically refers to news articles. I would recommend “publications”
- Line 49: Please capitalize “figure” i.e. Figure 1. Same on Line 162 and elsewhere.
- Line 88: Please use \citet{biderman2023}, not \citep{biderman2023}. The parentheses should not be around Biderman if Biderman is the subject in the sentence.
- Line 194: Which Llama 2 uses $d=4096$ and $K=32$? I believe you mean Llama 2 7B, but please check, and please be specific.
- Line 247: The brackets are facing outwards instead of inwards in “]0, 1[“
- Line 279: Please capitalize Theorems and Sections
- Line 314: Please capitalize Figure

Missing Citations (I’ll add more as they come to me)
- https://arxiv.org/abs/2406.11715

**Questions:**

N/A

---

> ### Author Response · Authors · 2024-11-23
> **Answer to jqv5**
>
> Dear Reviewer,
>
> Thank you for your valuable feedback. We have provided a global response to the four reviews we received. This message offers additional details addressing the specific concerns you raised.
>
> First, regarding the requirements you mentioned to improve our score:
> 1. You are absolutely right. We have changed the title and the abstract to communicate this aspect more prominently (see global answer).
> 2. Unfortunately, extractability measures such as k-memorization are not applicable to our settings. We have added a clear discussion on this point in lines 93–106 and 129–141. See the global answer for more details.
> 3. We re-ran all experiments with only 1 epoch of training.
> 4. We changed the narrative accordingly.
> 5. These three metrics are now jointly studied in Section 3.
> 6. Unfortunately, we were not able to formulate an interesting theoretical result for the loss or the logit gap. Indeed, these metrics depend heavily on the directions used by the classification head to compute the logits. Consequently, we would need to formulate hypotheses on the accuracy of the classification head, which is complicated because training is incomplete when we perform the measure. We did not need such hypotheses for PSMI because its definition considers an expected value over all possible directions.
> 7. Minor language suggestions you provided: thank you, we fixed all of them, except for the brackets in “]0, 1[”. Our demonstration requires that epsilon is not equal to 1.
>
> Besides these requirements, regarding the weaknesses you have highlighted:
> - In the related work, we have organized lines 129–141 to distinguish methods that are applicable to generative or discriminative models.
> - Section 3.1 indeed had weaknesses. We have removed it and replaced it with more factual discussions (see global answer).
> - We have removed Figure 4B and replaced it with more informative figures related to CIFAR-10.
> - The baseline of Biderman et al. is indeed not directly applicable to our setting. We have adapted our description accordingly (see lines 93-106, 1001–1017 and our global answer).
>
> Please let us know if our revision correctly addresses your concerns. We would be happy to discuss any aspect further.

---

> ### Comment · Reviewer_jqv5 · 2024-11-26
> **Response to Authors's Response (Round 1)**
>
> **Some Changes that I appreciated:**
>
> - Title, abstract and introduction are more compelling (in my opinion, at least)
> - New paragraph on lines 85-92
> - The PyPI package for an automated estimator of PSMI (at least, I don't remember this being included in the original submission)
>
> **Major Remaining Pieces:**
>
> - The title should not have any acronym "LLM" in it. Spell the term out, please.
> - Looking at the new manuscript that studies vision classifiers (Figure 5), I now realize (perhaps embarrassingly late) that this paper has little to do with language models and instead deals specifically with classifiers. Consequently, I feel like the narrative is now contorted. The focus on LLMs in the title, abstract, introduction, methods and results is misdirection. This paper is fundamentally about classifiers.
>
> **Minor Remaining Pieces:**
>
> - On Line 17, the antecedent of "it" is "the new approach" but this isn't exactly clear. Perhaps you could say "The method is efficient from the early stages of training and readily adaptable to other classification settings"
> - On Line 224, the authors didn't specify which Llama 2 model is being used with $d=4096$ and $K=32$. Ideally the authors could just say "Llama 2 7B"
> - Overall, the writing is still rough. I urge the authors to spend more time polishing.

---

> ### Author Response · Authors · 2024-11-26
> **Answer to jqv5 (Round 2)**
>
> Dear Reviewer,
>
> We are grateful for your feedback and appreciate that you took the time to re-examine our work, especially given the extensive modifications made following the first round of reviews. We have submitted a 2nd revision to address your concerns regarding our 1st revision.
>
> Thank you for highlighting the changes you found valuable. The release of the PyPI package is indeed an addition in the 1st revision. Based on your observation that PSMI could lead to implementation errors due to the lack of a public implementation, we decided to release a package in the near future that will be applicable even beyond the scope of this paper.
>
> Regarding the major revision that you highlighted:
> 1. We changed the title accordingly.
> 2. Memorization in a classification setting:
>    - We apologize if the motivation appeared misleading. This work initially emerged as a study of how memorization occurs in classification scenarios, with the aim of extending our understanding to future studies on the memorization of chunks of phrases by LLMs. Our intent was that using a transformer-based model in a text classification scenario could provide valuable insights for future research in this direction. Thus, the introduction was written with this spirit in mind and not to mislead readers.
>    - Additionally, as stated in lines 202–206, generative models are becoming ubiquitous for text-related tasks. They are increasingly being used to produce formatted outputs, effectively replacing traditional classification models. We felt that focusing solely on BERT-like classification models or WRNs trained on CIFAR-10 would deviate significantly from the SOTA approaches used in most practical scenarios. This is why we conducted most of our experiments on generative models, utilizing the NTP task to predict labels in a manner similar to how models are typically fine-tuned on instruction-response pairs.
>
> Regarding the minor revisions you mentioned:
> 1. Thank you, we fixed it.
> 2. We moved the mention "Llama 2 7B" from line 221 to line 224 for more clarity.
> 3. We have reworked the text to improve clarity, and hope that the quality of the 2nd revision meets your expectations.
>
> Please let us know if we correctly addressed your concerns. We would be happy to discuss any aspect further.

---

### Official Review · Reviewer_APAo · 2024-11-02

**Soundness:** 2
**Presentation:** 2
**Contribution:** 2
**Rating:** 3
**Confidence:** 3

**Summary:**

The paper studies memorization when fine-tuning LLMs for classification tasks by applying a metric called pointwise sliced mutual information (PSMI) from prior work. The authors show how PSMI can be used to detect which examples are prone to memorization early during training by stopping the training loop after the loss has decreased by 95%, computing PSMI of each dataset sample, and flagging samples with negative PSMI as potentially memorized. They provide theoretical evidence for this approach and empirically demonstrate its efficacy by training Mistral 7B, LLama 2 7B and Gemma 7B models on the MMLU, ETHICS and ARC benchmarks. The results show that PSMI is competitive with previous metrics and can be run more efficiently.

**Strengths:**

1. The paper empirically demonstrates that the PSMI metric is a strong indicator of memorization in settings where LLMs are used for classification, performing as well or better than alternative metrics.
2. PSMI is also shown to be computationally more efficient than alternative metrics.
3. The authors theoretically show that PSMI can be used to detect memorized outliers.
4. The arguments for using false positive rate (FPR) at high true positive rate (TPR) to evaluate the effectiveness of memorization metrics are convincing.

**Weaknesses:**

1. The applicability of the results is quite narrow, for a paper targeted at generative LLMs. The paper focuses on detecting memorization when fine-tuning LLMs for classification, i.e. to predict the label for an input sequence. However, most LLM fine-tuning settings are generative, i.e. with the goal of learning to generate a particular distribution of text. The methodology of the paper relies on categorical labels as targets, so it cannot be applied to generative settings. Therefore, it also cannot be used to mitigate regurgitation-based memorization issues where the model might e.g. regurgitate private or copyrighted information, that are primarily of concern for generative models. I think the PSMI metric proposed here would make more sense in typical classification settings, but feels out of place for generative models. The choice of fine-tuning models on datasets typically used for benchmarking (MMLU, ETHICS, ARC) also feels strange in this context.
2. The discussion in 3.1 about the dynamics of memorization and the three phases of "pattern initialization", "pattern complexification" and "pattern degeneration" seems rather speculative to me. For instance, in line 345 onwards, the authors claim that "increasingly complex patterns bring easy elements closer to the decision boundary". Is there any evidence for a dynamic like that? Further, in line 349 the authors state that the training loss further decreases during the degradation phase, but that is not really visible in Figure 2c). Finally, the text repeatedly refers to accuracy values which are not shown.
3. The paper uses LoRA to fine-tune models. LoRA limits the degree to which representations can change due to the low-rank deltas and limited contribution of the deltas (modulated by the $\alpha$ parameter) to the base model's representations. Therefore, it is possible that for non-LoRA fine-tuning, e.g. full fine-tuning, representations might change in a different way. Since the PSMI metric relies on the interplay between representations and labels, it might behave differently for other types of fine-tuning.
3. There are a number of minor language issues, e.g. "help the model to generalized to similar..." (l 37), "train one or more shadow model, which ..." (l 87), "they are likely to be difficult point ..." (l 356), "mutual information is null" -> zero (l 207), mixing past and present tense (e.g. "observe" and "observed"), somewhat non-standard lowercase notation for figure and other references, etc.

**Questions:**

1. What were the input samples on which memorization was detected? Were samples entire input rows from the datasets or were substrings also considered? While models might not memorize entire input sequences, they might memorize substrings, so this choice might lead to different degrees of measured memorization. Prior work on memorization in LLMs typically uses substrings [1, 2].
2. Appendix C.3 states that the evaluated model is only trained on 1% of the data, and the remaining 99% are used to train shadow models. Could the authors clarify how many samples the models are trained and evaluated over? Appendix C.3 also states that similar results are obtained when picking other data splits. How many splits do the authors evaluate? To better judge the reliability of the results, it would be very useful to annotate the plots with confidence estimates over the different runs.
3. In Figure 3, why does the Mahalanobis distance use representations from layer 13, but PSMI representations from layer 29?
4. Line 458 mentions additional results but does not mention where to find them. If those results are in the appendix it would be good to reference them there.

[1] Quantifying Memorization Across Neural Language Models, Carlini et al., https://arxiv.org/abs/2202.07646

[2] Memorization Without Overfitting: Analyzing the Training Dynamics of Large Language Models, Tirumala et al., https://arxiv.org/abs/2205.10770

---

> ### Author Response · Authors · 2024-11-23
> **Answer to APAo**
>
> Dear Reviewer,
>
> Thank you for your valuable feedback. We have provided a global response to the four reviews we received. This message offers additional details addressing the specific concerns you raised.
>
> Regarding the weaknesses you have highlighted:
> 1. [Applicability of the results] Your concern is valid, and our method is indeed only applicable to classification settings (see global answer). We have added justification for our choice of experimental settings (lines 199–215). Moreover, we ran experiments on a wide residual network trained from scratch on CIFAR-10 to demonstrate that our approach is applicable to a wide range of classification settings.
> 2. [Section 3.1] You are right; we have removed Section 3.1 and replaced it with more factual discussions (see global answer).
> 3. [Use of LoRA] This is indeed an important aspect of our experiments. To demonstrate that we do not rely on specific properties of LoRA, we applied our method as-is to a model trained from scratch, yielding conclusive results (see lines 451–485).
> 4. [Language issues] Thank you, we have fixed them.
>
> Regarding your questions:
> 1. We measure the memorization of entire samples, including both the full text and its label. Because we use membership inference attacks in a classification setting, we cannot apply extractability metrics, so we cannot measure memorization of substrings (see global answer).
> 2. Each model is trained on exactly 50% of the training samples, selected randomly. As a result, the 100 models we train in each setting have different training sets. Consequently, for every $k \in [0, 99]$, model number $k$ can be attacked using the 99 other models as shadow models for LiRA. See lines 792–842 for a formal description of this mechanism. Unfortunately, because we had to re-run all experiments with only 1 epoch of training, we did not have time to replicate our experiments using the other seeds. We argue that the reproducibility of our method is established by the diversity of settings we evaluate (4 different datasets and 4 different models for a total of 6 empirical settings).
> 3. We always compute the predictor at the layer for which it has the best performance. See Appendix D5 for an ablation study on the layers for Mahalanobis distance.
> 4. Thank you. In Section, 3, we have now referenced all additional results that are included in the Appendix.
>
> Please let us know if our revision correctly addresses your concerns. We would be happy to discuss any aspect further.

---

> ### Author Response · Authors · 2024-11-29
> **Further discussion would be greatly appreciated**
>
> Dear Reviewer APAo,
>
> Thank you very much for the time and effort you have dedicated to reviewing our paper. We greatly appreciate your constructive feedback. Given the extended discussion period, we believe that further correspondence with you would add significant value to our manuscript.
>
> We hope that our rebuttal has sufficiently addressed the concerns you raised. Please let us know if any questions or issues remain, as we would be more than happy to discuss them further.
>
> Best regards,
>
> The Authors

---

### Official Review · Reviewer_U5DN · 2024-11-04

**Soundness:** 2
**Presentation:** 3
**Contribution:** 2
**Rating:** 5
**Confidence:** 4

**Summary:**

The paper proses a measure to identify samples that are likely to be memorized using Pointwise Sliced Mutual Information (PSMI). The intuition is to use a model's snapshot from early in training when the model has learned "some" useful representations (but hasn't saturated to zero loss) to find possible outliers where PSMI between representation of an input X cannot predict its label Y. Such outliers are likely to be memorized as training proceeds, so they can be removed or dealt with as the model developers sees fit to ensure the final fully trained model doesn't suffer from pitfalls of memorization. Since the method uses a partially trained model very early on in training (typically 2-5% of training steps) and only requires forward passes through this partially model which makes it computationally much cheaper than training the full model and then removing memorized samples, after the fact. Empirical results show good FPR at a high TPR thus indicating the method could work in practice to detect a-priori the samples at risk of memorization.

**Strengths:**

- The method is simple and intuitive and isn't computationally very expensive
 - The paper is well written and properly compares the proposed method to baselines
 - I appreciated the section on ablations of hyperparameters

**Weaknesses:**

See questions

**Questions:**

- [Why is this approach specific to finetuning] In section 1.2 you mention that unlike Biderman et al [1] your approach is specific to finetuning. Why is this the case? Moreover, finetuning is typically much less computationally demanding than pretraining, thus I don't fully get why you put the same strict constraints on computational constraints as Biderman et al (who are focused on pretraining, thus these constraints make more sense in their setting).

- [Evaluation Metric] Why did you pick FPR at TPR = 0.75? This is a rather arbitrary number and would be rather context dependent. Why don't you just show the tradeoffs like you do in Fig 3a, or simply take the area under the curve if you must distill it into a number?

- [Sample ordering] What if the sample that is at risk of being memorized (ie low PSMI) has already been seen in the partial model? That is, the order in which you feed finetuning data will matter. Currently you are assuming any at the risk of memorization will not have been seen by the partial model you use to actually find the samples with low PSMI. But if any sensitive information has already been seen by the partial model then it seems like

 - [Finetuning loss] What kind of finetuning are you doing? I could not find any information on this anywhere in the paper. Are you finetuning on the benchmarks like MMLU to literally to classification by attaching a classification head? Or are you doing supervised finetuning where you mask out the loss of question?

 - [Why is MIA an indication of memorization] I am confused by the choice of using MIA (LiRA) to infer ground truth memorization. Since you have full access to finetuning data why don't you use standard memorization measures like in [1,2] to infer what has or has not been memorized? MIA also has its own concerns, see [3].

 - [No improvement over logit gap or loss baselines] I appreciate that the authors compared fairly with baselines, however I do not see any claimed improvements on the FPR vs TPR tradeoff. Authors say that PSMI requires no tuning of threshold, which may be true but PSMI requires tuning, arguably more expensive hyperparams like when during finetuning should one stop (currently chosen as 95% in Algorithm 1) or which layer to use for PSMI (currently chosen as final layer in Algorithm 1).

 - [Need more experiments to confirm broad applicability] The experiments in the paper are good, however to support the claim that this method is broadly applicable there is a need to test this on more kinds of finetuning methods. Current evals are only on LoRA finetuning which can have its own effects on memorization [4]. I would expect comparison with full finetuning as well as a few other popular parameter efficient methods like bitfit, softprompts etc.

 - [Minor concern] You say in related work that "However, this belief was challenged by Zhang et al. (2017), who proved that a model can simultaneously perfectly memorize random labels and achieve state-of-the-art generalization." -- however you can see Figure 1 in Zhang et al and this is not what they claim. They in fact show that a model can perfectly memorize random label samples but such models totally fail to generalize.


[1] Biderman et al. https://arxiv.org/pdf/2304.11158

[2] Carlini et al. https://arxiv.org/abs/2202.07646

[3] Duan et al. https://arxiv.org/pdf/2402.07841

[4] Biderman et al. https://arxiv.org/pdf/2405.09673

---

> ### Author Response · Authors · 2024-11-23
> **Answer to U5DN**
>
> Dear Reviewer,
>
> Thank you for your valuable feedback. We have provided a global response to the four reviews we received. This message offers additional details addressing the specific concerns you raised.
>
> Regarding the weaknesses you have highlighted:
> 1. [Why is this approach specific to finetuning] You are right, our approach is not specific to fine-tuning. We have clarified this aspect in lines 199–215 and conducted experiments on a model trained from scratch on CIFAR-10 (see global answer). Moreover, we set this strict constraint of predicting memorization before the end of training because (1) the practitioner does not have the budget to compute ground-truth memorization anyway, and (2) for researchers aiming to develop empirical defenses focused on the most vulnerable samples, it is important to detect them before they are memorized.
> 2. [Evaluation Metric] This is a valid point. We now use AUC instead of FPR@TPR=75% (see global answer).
> 3. [Sample ordering] This is indeed a valid concern, and we agree that a practitioner using our method to implement a mitigation strategy should take this into account. However, we believe this concern falls outside the scope of our paper. As mentioned in lines 183–184, we focus solely on developing a reliable predictor of memorization. Since prediction is made before the end of epoch 1, it is necessary that some samples were used while others were not. Deciding how to handle samples that have already been processed depends heavily on the concrete application scenario.
> 4. [Finetuning loss] We are using LoRA fine-tuning. We use a multi-choice question template and ask the model to predict the number of the correct answer using next-token prediction task. The loss is only computed for the last token. See lines 211–215.
> 5. [Why is MIA an indication of memorization] Unfortunately, these measures are not applicable to our classification setting, which is why we resorted to MIA (see global answer). The concern you raised regarding the applicability of MIA to LLMs is valid. However, the paper you mentioned only raises concerns about cheap MIAs that avoid training shadow models, such as the loss of zlib. They do not evaluate strong MIA that involve shadow models (such as LiRA) due to the computational cost involved.
> 6. [No improvement over logit gap or loss baselines] You are absolutely right. We have modified the narrative to communicate that all three methods can be used and outperform the baseline (see global answer).
> 7. [Need more experiments to confirm broad applicability] We have run experiments on a wide residual network trained from scratch on CIFAR-10. Because this setting differs significantly from the fine-tuned LLMs we have studied so far, we believe that our method is applicable to a wide range of classification scenarios (see global answer).
> 8. [Minor concern] Thank you, we fixed it (see lines 144–145).
>
> Please let us know if our revision correctly addresses your concerns. We would be happy to discuss any aspect further.

---

> ### Author Response · Authors · 2024-11-29
> **Further discussion would be greatly appreciated**
>
> Dear Reviewer U5DN,
>
> Thank you very much for the time and effort you have dedicated to reviewing our paper. We greatly appreciate your constructive feedback. Given the extended discussion period, we believe that further correspondence with you would add significant value to our manuscript.
>
> We hope that our rebuttal has sufficiently addressed the concerns you raised. Please let us know if any questions or issues remain, as we would be more than happy to discuss them further.
>
> Best regards,
>
> The Authors

---

### Author Response · Authors · 2024-11-23
**Common answer to all reviewers**

Dear Reviewers,

Firstly, we express our gratitude for your insightful feedbacks. Your comments have accurately highlighted some weaknesses in our paper, which greatly aids in refining it. Following your precise recommendations, we have made substantial modifications to the original version of our manuscript. We believe that this revised version significantly enhances the clarity of our approach and adequately addresses the concerns you raised.

To address your review, we summarize below the most substantial modifications we have made. The line numbers provided correspond to the revised version.

## 1- Scope of our Paper

As you correctly pointed out, our method is only applicable to models trained for a classifiction task. While this does limit the range of potential applications, this area of research remains active [1, 2, 3]. Additionally, although our method is not applicable to LLMs trained for generative tasks, it is fully applicable to any model trained for classification, including vision models. To demonstrate this, we applied our method without modification to a wide residual network trained from scratch on CIFAR-10, achieving very promising results. We believe that the excellent performance observed in this setting, which differs substantially from the fine-tuning of LLMs, demonstrates the broad applicability of our approach to a variety of classification scenarios.

**Main modifications to address your concerns:**
- We revised the title and abstract of the article to more clearly specify the application scenario.
- Lines 199–215: We added a discussion to justify our choice.
- Lines 451–488: We included experiments on CIFAR-10.

## 2- Definition of Memorization and Applicability of the Baseline

As you noted, we use vulnerability to a membership inference attack called LiRA as our ground-truth measure of memorization. This approach differs significantly from *extractability* measures, such as k-memorization used by [4]. Extractability is rarely employed for discriminative models because extraction/reconstruction attacks are complex to implement on these models and often yield lower performance. Consequently, measures like LiRA or counterfactual memorization are generally preferred in such settings.

These standard memorization measures in classification settings are computationally expensive due to the shadow models involved. This makes them impractical for practitioners operating under our threat model, who typically lack the resources to compute them. Moreover, the baseline proposed by [4] is also inapplicable under our threat model because measuring partial memorization is prohibitively costly. Nevertheless, we included it (after adaptation to this classification setting) because it is the only comparable approach we are aware of.

**Main modifications to address your concerns:**
- Lines 93–106: We clarify the applicability of the baseline from [4].
- Lines 129–141: We differentiate between methods applicable to discriminative and generative models in the literature.
- Lines 169–176: In our threat model, we explicitly state that practitioners cannot compute any ground truth measure of memorization.
- Lines 1001–1017: We provide details on the adaptation of [4] to our classification setting.

## 3- Similar Performance of PSMI, Loss, and Logit Gap

You pointed out that PSMI does not yield significantly better results than the loss and the logit gap. We believe the primary advantage of PSMI is that it allows practitioners to avoid selecting a threshold to distinguish memorized from non-memorized samples, thanks to Theorem 1. However, we agree that we should more prominently highlight that all three metrics can be effectively used and outperform the only comparable baseline we are aware of.

**Main modifications to address your concerns:**
- We revised the narrative throughout the paper to emphasize that all three metrics achieve significantly better results than the baseline.
- Lines 300–304: We explicitly mention that the loss can be used as an alternative to PSMI and refer to a new algorithm introduced in lines 1061–1066.

### [Parts 4, 5, and references in next comment]

---

> ### Author Response · Authors · 2024-11-23
> **Parts 4, 5 and references of the official comment**
>
> ## 4- Changing Experiments: Only 1 Epoch of Training, and AUC Instead of FPR@TPR=75%
>
> You noted that we trained our models for 10 epochs, leading to overtraining. Additionally, you pointed out that the FPR@TPR=75% metric is highly sensitive to the threshold set on the TPR. We agree with these observations and have made the appropriate adjustments in our paper.
>
> These changes have significantly altered our article, as they impact all the experiments we conducted. However, we believe they considerably improve the robustness of our results.
>
> **Main modifications to address your concerns:**
> - We re-ran all experiments using 1 epoch instead of 10.
> - We now consistently use AUC instead of FPR@TPR=75%. While FPR@TPR=75% aligns more closely with our threat model, avoiding an arbitrary threshold is of greater importance.
> - Lines 355–357: We updated the CPU/GPU consumption details accordingly.
>
> ## 5- Removing Section 3.1 to replace it with a more factual description
>
> Our discussion in this section indeed had weaknesses and lacked quantitative arguments. Moreover, it diverged from the core of our paper, which focuses on predicting memorized samples. Consequently, we removed Section 3.1 and replaced it with a more factual description of our measures in the ablation studies.
>
> **Main modifications to address your concerns:**
> - We have added a more factual description in Figure 3 and lines 397–407.
> - Additionally, we have plotted detailed measures throughout training (loss for both train and test, accuracy for both train and test, performance of the predictors, PSMI, and memorization) in Appendix D2 (lines 1128–1187).
>
> ## Conclusion
>
> In addition to this global response, we will provide individual replies addressing the specific concerns raised by each reviewer. We apologize for the delay in our response, which was due to the substantial time required to re-run experiments with 1 epoch and implement our method with CIFAR-10. We hope that you will have sufficient time to re-examine our paper, and we would be happy to discuss any aspect further.
>
> ## References
>
> [1] Aerni et al. Evaluations of Machine Learning Privacy Defenses are Misleading. 2024.
>
> [2] Mireshghallah et al. Quantifying Privacy Risks of Masked Language Models Using Membership Inference Attacks. 2022.
>
> [3] Carlini et al. The Privacy Onion Effect: Memorization is Relative. 2022.
>
> [4] Biderman et al. Emergent and Predictable Memorization in Large Language Models. 2024.

---

> ### Author Response · Authors · 2024-11-23
> **Provide Latexdiff**
>
> Dear Reviewers,
>
> We have observed that the OpenReview tool for comparing PDF versions fails to correctly compare our revisions. To accommodate your review, we provide the following `latexdiff`, available from this anonymous GitHub account that we created for this purpose:
>
> https://github.com/user7493131606/latexdiff/blob/8d076e916a0562bf1a6133e487eb607e012a4c72/Latexdiff_ICLR2025_paper_10739.pdf

---

### Author Response · Authors · 2024-11-26
**Upcoming deadline + update to accommodate changes required by reviewer jqv5**

Dear reviewers,

Thank you for the time and effort you have dedicated to reviewing our submission. We have carefully considered your feedback and made revisions accordingly (see "Common answer to all reviewers" comment above, submitted on Nov 23rd). As the deadline for submitting the updated paper is tomorrow, we kindly ask if you could review the revised version and provide any additional comments, if possible. We greatly appreciate your support and understanding.

Along with this message, we are submitting a new revision incorporating the changes requested by reviewer jqv5 following our first revision. These changes primarily focus on refining the writing of the article. To help with your review, we provide a latexdiff comparing our first revision (submitted on Nov 23rd) with this revision. You can access it through the following anonymous GitHub repository:

https://github.com/user7493131606/latexdiff/blob/8425ef5d779f4202594906b2033b177651ff969a/Latexdiff_ICLR2025_paper_10739_second_revision.pdf

Best regards,
The Authors

---

### Meta-Review · Area_Chair_PQWh · 2024-12-19

**Metareview:**

The authors study memorization in fine-tuned classifiers using PSMI. The core idea of the paper is to use early checkpoints where we might be able to identify (both cheaply and reliably) points that are likely to be memorized by the full model. To identify these points, the authors look for outlier-like points that will likely be memorized.

The paper as submitted had some fairly severe issues - most of which are pointed out by reviewer jqv5 - including the fact that this is really not a LLM specific memorization paper and does not target most of the sequence memorization settings that are of interest in that literature, and the experiments used settings (e.g. 10 epochs) that were unrealistic for the LLM setting.

The rebuttal process did change the paper quite a bit, but as the reviewers note - this is really still not an LLM memorization paper and really should be reviewed alongside the fairly large existing literature on MIA and memorization in deep neural networks broadly.

**Additional Comments On Reviewer Discussion:**

There was extensive back and forth on reviewer discussion ,and the authors provided many updates that engaged with the reviewers. While I think this was great for improving the quality of the paper overall, and dealing with some glaring issues (like the 10 epoch thing) I think the broader issues about the positioning of this work remain -- importantly, if the positioning of this work was to be more general to any classifier memorization, I think we should get a different reviewer pool and the reviewers would ask for a very different set of baseline type comparisons. In that sense, I think many of the issues remain unaddressed.

---

### Decision · Program_Chairs · 2025-01-22

Reject